# Genetic accommodation via modified endocrine signalling explains phenotypic divergence among spadefoot toad species

Saurabh S. Kulkarni[1,5], Robert J. Denver[2,3], Ivan Gomez-Mestre[4] & Daniel R. Buchholz[1]

Phenotypic differences among species may evolve through genetic accommodation, but mechanisms accounting for this process are poorly understood. Here we compare hormonal variation underlying differences in the timing of metamorphosis among three spadefoot toads with different larval periods and responsiveness to pond drying. We find that, in response to pond drying, *Pelobates cultripes* and *Spea multiplicata* accelerate metamorphosis, increase standard metabolic rate (SMR), and elevate whole-body content of thyroid hormone (the primary morphogen controlling metamorphosis) and corticosterone (a stress hormone acting synergistically with thyroid hormone to accelerate metamorphosis). In contrast, *Scaphiopus couchii* has the shortest larval period, highest whole-body thyroid hormone and corticosterone content, and highest SMR, and these trait values are least affected by pond drying among the three species. Our findings support that the atypically rapid and canalized development of *S. couchii* evolved by genetic accommodation of endocrine pathways controlling metamorphosis, showing how phenotypic plasticity within species may evolve into trait variation among species.

[1] Department of Biological Sciences, University of Cincinnati, Cincinnati, OH 45221, USA. [2] Department of Molecular, Cellular and Developmental Biology (MCDB), University of Michigan, Ann Arbor, MI 48109, USA. [3] Department of Ecology and Evolutionary Biology (EEB), University of Michigan, Ann Arbor, MI 48109, USA. [4] Ecology, Evolution and Development Group, Doñana Biological Station, CSIC, Almonte E-41092, Spain. [5] Present address: Department of Pediatrics, Yale School of Medicine, New Haven, CT 06520, USA. Correspondence and requests for materials should be addressed to I.G.-M. (email: igmestre@ebd.csic.es) or to D.R.B. (email: buchhodr@ucmail.uc.edu)

Phenotypic plasticity is a common property of developing organisms, where a single genotype can give rise to different phenotypes in different environments[1]. Theoretical, empirical, and laboratory selection approaches indicate that divergent selection, acting on plastic traits, can quickly cause genetic changes in the regulation, expression, and/or frequency of traits among populations within a species, a process known as genetic accommodation[1, 2]. The possibility that phenotypic differences among species may stem directly from genetic accommodation[1, 3, 4] is suggested by examples where differences among species parallel environmentally induced differences among individuals within a related species[3–6]. However, how mechanisms of trait regulation evolve during genetic accommodation is poorly understood and of vital importance for elucidating why/how lineages (populations or species) differ in phenotype.

Acceleration of metamorphosis occurs in response to pond drying in many anuran amphibians and is largely dependent upon increased production and action of thyroid hormone (TH; the primary morphogen controlling amphibian metamorphosis) and corticosterone (CORT; a stress hormone that synergizes with TH to promote tissue transformation)[7–10]. Exposure to pond drying leads to activation of hypothalamic corticotropin-releasing hormone (CRH) neurons, and CRH stimulates secretion of adrenocorticotropic hormone (ACTH) and thyroid stimulating hormone (TSH) by the pituitary gland. Circulating ACTH acts on the interrenal glands and TSH acts on the thyroid glands to stimulate secretion of CORT and TH, respectively. Thus, because treatment with exogenous TH and CORT is known to accelerate metamorphosis in frogs, marked elevation of endogenous TH and CORT levels explains at least in part the developmental acceleration of tadpoles exposed to pond drying[7].

Extant spadefoot toad species occupy a range of habitats from arid to semi-arid to deciduous temperate forest and lay eggs in bodies of water with different degrees of permanence, from ephemeral pools to long-lived seasonal ponds[11]. Adaptations to relative pond permanence are seen in the tadpoles of different spadefoot toad species[12, 13]. Couch's spadefoot toad (*Scaphiopus couchii*) lays eggs in the most ephemeral pools and has the shortest larval period known of all anurans. The Western spadefoot toad (*Pelobates cultripes*) lays eggs in long lasting pools and has the longest larval period of the three species studied, and the New Mexico spadefoot toad (*Spea multiplicata*) is intermediate in duration of breeding ponds and in its larval period[12, 14]. Accelerated metamorphosis and increased TH and CORT content in response to pond drying have been observed in the spadefoot toad species, *Spea hammondii* (a relative of *S. multiplicata*) and *P. cultripes*[9, 15]. Further, individuals undergoing stress-induced metamorphosis exhibit morphological consequences of the increased hormone contents, including smaller body size, shorter relative limb length, and reduced levels of abdominal fat[16, 17]. These morphological changes can also be induced by exogenous treatment with TH and CORT[18, 19].

A key observation is that even though *S. couchii* was described as an example of developmental plasticity in previous literature[20, 21], this species has dramatically reduced plasticity compared to *P. cultripes*[16, 17]. Furthermore, in comparison to the other spadefoot species, *S. couchii* has reduced size at metamorphosis, shorter hind limbs, altered head shape, altered timing of gonad differentiation, and smaller abdominal fat bodies[10, 13, 16, 17]. These morphological traits are constitutively expressed in *S. couchii* but observed in *P. cultripes* and *S. multiplicata* only when they are exposed to low water levels. Thus, phenotypic differences among species in traits associated with the larval period mirror within-species differences caused by changes in developmental rate induced by pond drying and exogenous hormones. Further, based on phylogenetic character state reconstruction, plasticity in duration of the larval period is ancestral among extant spadefoot toad species, and rapid development in *S. couchii* is derived[16, 22]. We therefore hypothesize that species differences in larval period and associated traits evolved through genetic accommodation via altered endocrine regulation.

To test this hypothesis with respect to endocrine regulation, we investigated the mechanistic basis for phenotypic differences among spadefoot toad species that differ in the duration of their larval period and their degree of plasticity in the timing of metamorphosis. We raised tadpoles of three species (*S. couchii*, *P. cultripes* and *S. multiplicata*) in the laboratory, exposed them to simulated pond drying, and measured whole-body TH and CORT content and standard metabolic rate (SMR). We show that species differences in larval period and responsiveness to pond drying are associated with modification of endocrine signaling pathways that control tadpole metamorphosis. Our findings support the hypothesis that altered endocrine signalling from genetic accommodation of larval period duration accounts for the phenotypic differences observed among spadefoot toad species.

## Results

**Larval period and plasticity differences among species.** The evolution of differences in larval period and developmental plasticity among spadefoot toad species provides a clear example of an evolutionary pattern consistent with genetic accommodation (Fig. 1)[16, 17, 23]. In particular, *P. cultripes* exhibits significant plasticity in the timing of metamorphosis, as its larval period was shortened by ~40% when exposed to low water conditions (31.4 ± 0.56 [mean ± standard error] days in high water vs. 18.4 ± 0.54 days in low water) (Fig. 1b)[13, 16, 17]. By contrast, tadpoles of *S. couchii*, which breed in ephemeral ponds, had a much shorter larval period (6.6 ± 0.16 days in high water) and did not accelerate development when exposed to simulated pond drying (Fig. 1b). The larval period of *S. multiplicata* was intermediate to *S. couchii* and *P. cultripes* and also showed an intermediate plastic response to simulated pond drying (the larval period shortened by ~20%; 9.1 ± 0.28 days in high water vs. 7.2 ± 0.36 days in low water) (Fig. 1b)[13, 16, 17]. The short larval period with reduced plasticity of *S. couchii* represents a derived and canalized developmental state[16, 22]. The divergence of reaction norms among spadefoot toad species is consistent with evolution by genetic accommodation of the timing of metamorphosis driven by selection in ponds with different hydroperiods[13, 16, 17].

**Differential endocrine responses to simulated pond drying.** To investigate a mechanistic (i.e., endocrine) basis for differences among species in developmental timing, we measured whole-body TH and CORT content under high water and low water conditions in tadpoles of the three species. We found that, when matched for developmental stage and raised in high water environments, *S. couchii* had significantly higher whole-body TH content compared with *P. cultripes* (3.69 fold difference; $F_{1,65} = 14.0$, $p = 0.00039$; Fig. 1c), which is consistent with our previous observations[10]. We were unable to obtain measures of TH for *S. multiplicata*. Furthermore, *P. cultripes* tadpoles significantly increased TH in response to decreased water level ($F_{1,30} = 18.14$, $p = 0.00019$), whereas TH content remained unchanged in *S. couchii* tadpoles regardless of water level ($F_{1,35} = 0.15$, $p = 0.71$; Fig. 1c).

As was the case for TH, *S. couchii* had higher whole-body CORT content compared with *P. cultripes* and *S. multiplicata* in the high water level treatment ($F_{2,62} = 6.12$, $p = 0.0038$) (Fig. 1d). This difference was seen in early prometamorphic (Gosner stage 35) and late prometamorphic (Gosner stage 38) tadpoles, but disappeared at metamorphic climax (Gosner stage 42; Fig. 2a). Similar to the increase in TH in *P. cultripes* after simulated pond drying, both *P. cultripes* ($F_{1,15} = 12.83$, $p = 0.0027$) and

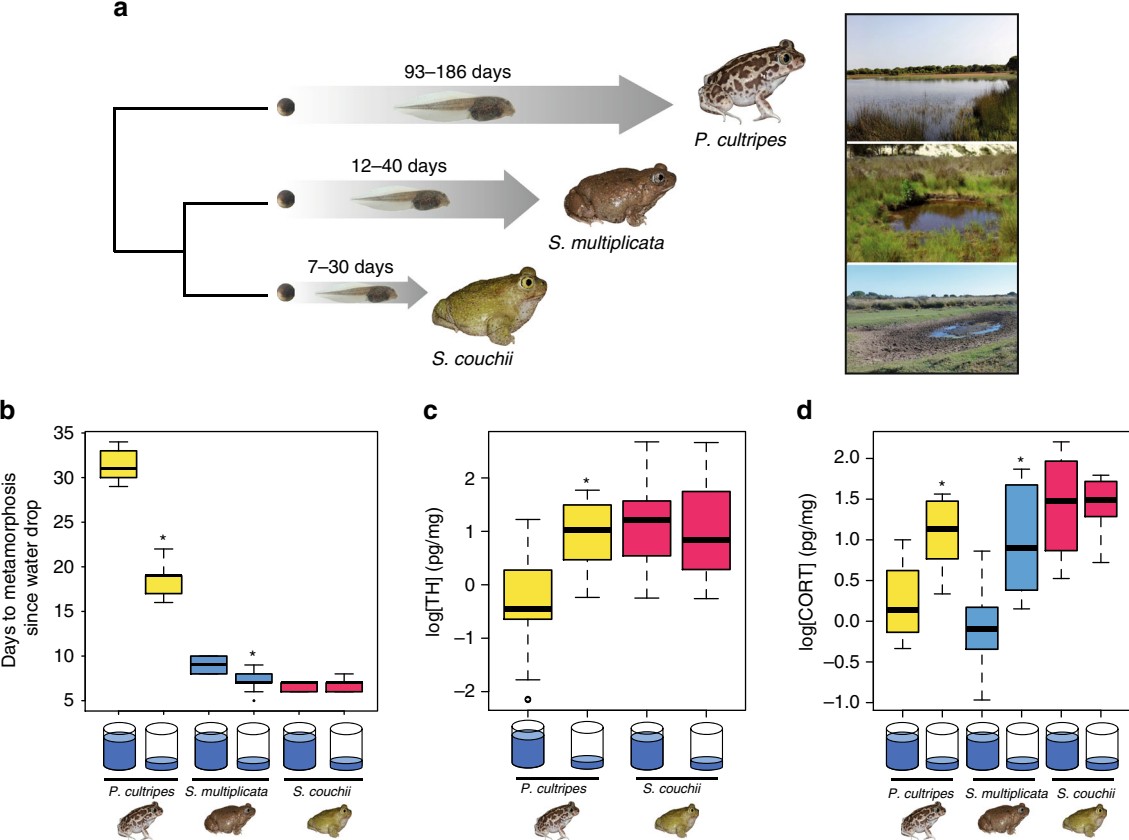

**Fig. 1** Endocrine basis of species differences in larval period and plasticity. **a** Spadefoot toads have evolved the largest differences in development rate among anurans. Old World spadefoot toads (*Pelobates* spp.) tend to breed in long lasting ponds and have a relatively longer larval period. The closely related New World species have evolved faster developmental rates, and *S. couchii* in particular shows the fastest developmental rate known in anurans. Phylogenetic relationships among spadefoot genera with corresponding larval period durations and a typical pond for each genus are shown. **b** Simulated pond drying initiated at Gosner stage 35 (early prometamorphosis) and maintained until measurement at Gosner stage 42 (metamorphic climax) induced marked developmental acceleration in *P. cultripes* (in yellow), an intermediate response in *S. multiplicata* (in blue), and no significant response in *S. couchii* (in red). Boxes in the boxplots indicate the median and the upper and lower quartiles, whereas the whiskers indicate the minimum and maximum values excluding outliers. Sample size was 10 for all species and treatments. **c**, **d** Developmental acceleration in *P. cultripes* in response to simulated pond drying is regulated by **c** increased thyroid hormone (TH; $p = 0.00019$) and **d** increased corticosterone (CORT; $p = 0.0027$). We observed a similar degree of CORT increase regulating adaptive plastic responses in *S. multiplicata* (**d**; $p = 0.00083$). However, *S. couchii* exhibited a higher and invariant tissue content of both hormones **c**, **d** underlying its minimal adaptive developmental plasticity. Simulated pond drying was initiated at Gosner stage 35 (early prometamorphosis), and hormone tissue contents were assessed using tadpoles at Gosner stage 38 (late prometamorphosis). Sample sizes were 7 and 9 for TH in *P. cultripes* in high and low water levels, and 10 for *S. couchii* for both treatments. Sample sizes for CORT data were 9 and 8 for high and low water levels in *P. cultripes* and 10 for the other species and treatment combinations. Asterisks indicate statistically significant differences between tadpoles within a species raised in high vs. low water. Photo credits: I. Gomez-Mestre

*S. multiplicata* ($F_{1,18} = 16.04$, $p = 0.00083$) had elevated whole-body CORT content after exposure to low water level; whereas decreased water level did not alter CORT content in *S. couchii* ($F_{1,18} = 0.004$, $p = 0.953$) (Fig. 1d). These differences in hormone content in response to low water among these species reflects the fact that *P. cultripes* and *S. multiplicata* accelerate development in response to low water but *S. couchii* does not.

**Differences in SMR among species.** We also compared SMR among the three spadefoot toad species, because our previous studies supported that developmental acceleration entailed a higher metabolic cost[9, 17]. Tadpoles that accelerated development displayed reduced abdominal fat[17], and higher measures of oxidative stress, likely due to increased lipid catabolism[9]. Such results would further support the pattern of genetic accommodation where the metabolic state observed in plastic species undergoing accelerated development in drying ponds is commensurate with that of the species with constitutively rapid

development. We found a positive correlation between SMR and developmental rate (Fig. 2b). In particular, after controlling for differences in body size, the SMR of the fast-developing *S. couchii* was 2.9-fold higher than that of *S. multiplicata* and 10.9-fold higher than that of *P. cultripes* at early prometamorphosis (Gosner stage 35) (Fig. 2b). Similar results were obtained in late prometamorphosis (Gosner stage 38) and metamorphic climax (Gosner stage 42) (Fig. 2b).

**Differential effects of CORT manipulation on SMR.** The differences in SMR among species during prometamorphosis paralleled their differences in whole-body CORT content. Because there is evidence that SMR can be directly increased by exogenous CORT in other amphibians like red-legged salamanders (*Plethodon shermani*)[24], we investigated whether experimentally manipulating CORT would influence SMR in spadefoot toads. We found that the SMR of *P. cultripes* and *S. multiplicata* was increased after treatment with CORT ($p = 0.008$ and $p = 0.013$,

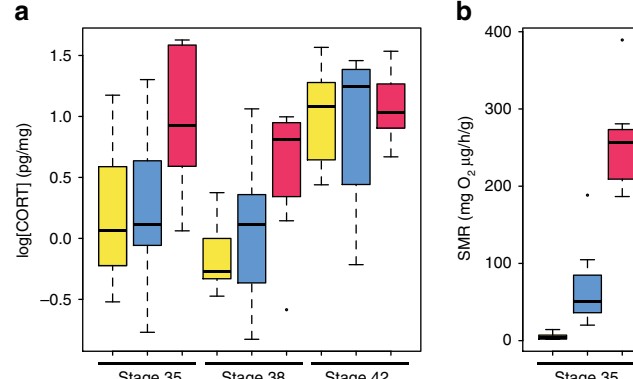

**Fig. 2** Hormonal and metabolic divergence among species across developmental stages. **a** Whole-body corticosterone (CORT) content was measured at the initiation of metamorphosis (Gosner stage 35), mid-metamorphosis (Gosner stage 38), and climax of metamorphosis (Gosner stage 42) for each spadefoot toad species under high water volume conditions. CORT contents were maximal and similar at metamorphic climax among species, but *S. couchii* had high, invariant CORT contents throughout the larval period unlike its relatives with longer and more plastic larval periods. **b** Size-corrected standard metabolic rate (SMR) was determined across the same developmental stages among species. SMR was fairly constant across development for each species, but the fastest developing species, *S. couchii*, showed the highest SMR, whereas the slowest developing species, *P. cultripes*, had the lowest SMR and *S. multiplicata* had an intermediate SMR. Sample sizes were 9, 11, and 10 for *P. cultripes*, 10, 10, and 9 for *S. multiplicata*, and 9, 5, and 8 for *S. couchii* at Gosner stages 35, 38, and 42

respectively, comparing high water vs. high water plus CORT) or exposure to low water ($p = 0.009$ and $p = 0.003$, respectively, comparing high water vs. low water) (Fig. 3a). Furthermore, the increase in SMR caused by the low water treatment could be reduced (*P. cultripes*) or blocked (*S. multiplicata*) by metyrapone, which inhibits CORT synthesis (Fig. 3a). The pattern of change in SMR matched the whole-body CORT content in *P. cultripes* and *S. multiplicata*, where animals from the CORT and low water treatments had significantly higher CORT content than the high water treatment. Also, tadpoles from the metyrapone treatment had a partially blocked low water-induced surge in CORT (Fig. 3b). In contrast, SMR of *S. couchii* was higher than in the other two species and was unchanged by any of the experimental manipulations ($F_{3,36} = 0.08$, $P = 0.97$) (Fig. 3a). Exogenous CORT, but not the low water treatment, increased CORT content in *S. couchii*, and CORT content was unaffected by metyrapone (Fig. 3b). Studies by others showed that non-toxic doses of metyrapone are not always effective at statistically significantly reducing CORT content, but expected CORT-related morphological and physiological effects may still be observed[25–27]. Thus, these results show that SMR is affected by manipulating CORT content, either through exposure to low water or directly by exogenous CORT addition in *P. cultripes* and *S. multiplicata*, and that high and invariant SMR in *S. couchii* is consistent with its constitutive rapid developmental rate.

## Discussion

Here we provide evidence that genetic accommodation of plasticity in the timing of metamorphosis of spadefoot toad species is explained by modification of endocrine signaling pathways. Our key finding is that while species with strong developmental responses to pond drying have corresponding endocrine responses, a species with short larval period has minimal developmental responses (i.e. canalized development) and its endocrine regulation of development reflects an intensified state of one end of the ancestral plasticity. We also found that tadpoles of *P. cultripes* and *S. multiplicata* increased their SMR in response to pond drying; whereas, *S. couchii* tadpoles had a constitutively high SMR that was unaltered by pond drying. The fact that the endocrine and metabolic adjustments to accelerate development in response to pond drying within plastic species (*P. cultripes* and *S. multiplicata*) corresponds to the canalized hormonal state of the larval period in *S. couchii* supports the hypothesis that genetic accommodation is

the evolutionary process relating ancestral developmental plasticity to phenotypic divergence among spadefoot toad species.

Genetic accommodation results from natural selection acting on trait regulation (in the present case, endocrine control of postembryonic developmental timing)[1]. To our knowledge, no studies have quantified selection on larval period in spadefoot toads, but larval period varies among *S. couchii* sibships, revealing genetic variation upon which selection can act[28]. Also, the strength of selection in favor of reduced larval period likely differs among species. That is, *P. cultripes*, whose larval period ranges from 93 to 186 days, breeds in long lasting temporary ponds that eventually dry up in summer; whereas, *S. couchii*, whose larval period ranges 7–30 days lays eggs in ephemeral desert pools that often dry in < 2 weeks[28, 29]. Furthermore, despite substantial overlap in breeding ponds, *S. couchii* often chooses pools that are too ephemeral for *S. multiplicata*, whose larval period ranges from 12 to 40 days. The exploitation of extremely ephemeral pools by *S. couchii* represents recurring episodes of stronger selection for rapid development than that experienced by *S. multiplicata*. Pond duration, rather than terrain aridity, is the most likely environmental factor that has driven the evolution of developmental rate in these species[22, 30].

We envision the following evolutionary scenario relating plasticity to phenotypic divergence among spadefoot toad species. First, Old and New World spadefoot toad clades (*Pelobates* vs. *Spea*/*Scaphiopus*) diverged at least by the Early Cretaceous[30–32], perhaps due to the formation of the Atlantic Ocean 150 million years ago. Plasticity in larval period was present before this split, as it is a common feature of amphibian development, and was retained in both lineages after the split[16]. Then, ancestors of *Scaphiopus* and *Spea* gradually experienced increased pond ephemerality compared with ancestors of *Pelobates*, but developmental plasticity of their larvae would have enabled them to persist. Because the timing of metamorphosis depends on endocrine signaling, continued selection for short larval periods to survive ephemeral ponds in New World spadefoot toad ancestors resulted in genetic changes that altered pathways controlling hormone production and/or action.

Endocrine regulation of tadpole metamorphosis is complex, and so numerous changes in the endocrine system could have accounted for the short larval period of *S. couchii*. However, the specific endocrine regulation we observed underlying canalized development in *S. couchii* compared to its relatives implicates

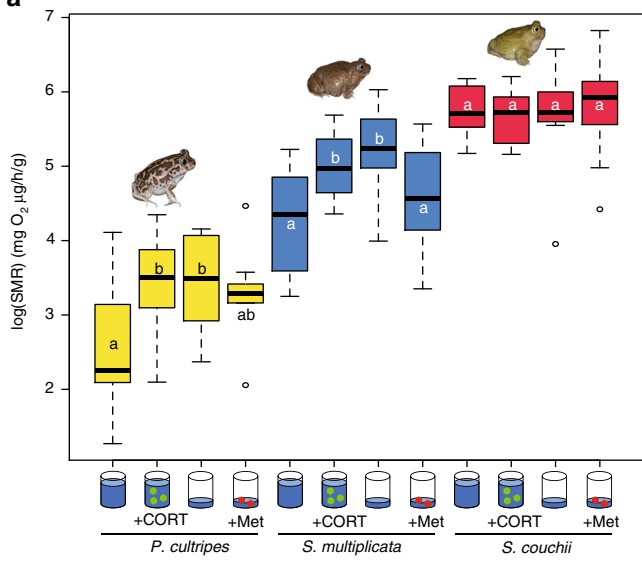

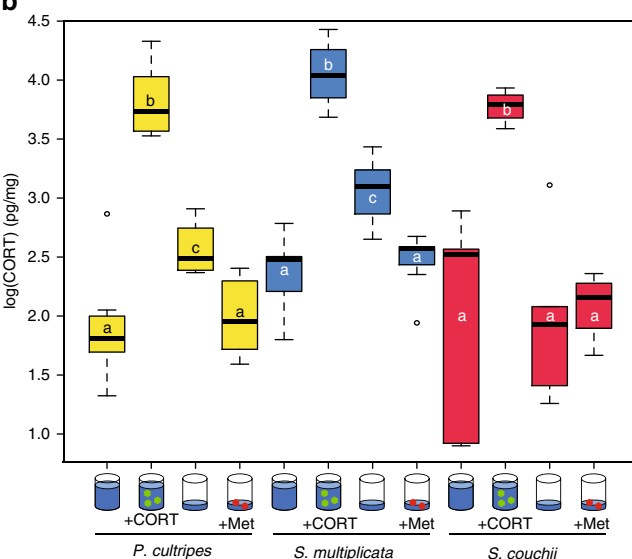

**Fig. 3** Hormonal and metabolic divergence among species across treatments. **a** Premetamorphic tadpoles of each spadefoot species were reared in high or low water level with exogenous CORT or metyrapone (Met, a CORT synthesis inhibitor) as shown. Species showing adaptive developmental acceleration in response to pond drying (*P. cultripes* and *S. multiplicata*) experienced steep increases in size-adjusted SMR when faced with reduced water level, which could be mimicked by treatment with exogenous CORT in high water and impaired by treatment with metyrapone in low water. In contrast, SMR in *S. couchii* was unresponsive to reduced water levels or even to exogenous CORT, hence showing a canalized high level of metabolism. **b** Whole-body CORT content was measured after 24 h in each treatment for each species. As expected, CORT content increased after low water treatment in *P. cultripes* and *S. multiplicata*, and exogenous hormone treatment increased CORT content in all species. Importantly, metyrapone treatment blocked the low water-induced increase in CORT content in *P. cultripes* and *S. multiplicata*, but metyrapone did not reduce the constitutive CORT content in *S. couchii*. Letters inside the bar graphs indicate significance groups within species based on post hoc significance tests. Sample sizes were as follows for the High, High+CORT, Low, Low+Met treatmentst: *P. cultripes* (10, 11, 10, 9), *S. multiplicata* (10, 10, 8, 10), and *S. couchii* (10, 10, 10, 10)

evolution by genetic accommodation. In particular, the endocrine changes and morphological consequences that accompany accelerated development in *P. cultripes*, such as increased TH and CORT, smaller body size at metamorphosis, shorter hind limb length, and reduced size of the abdominal fat bodies, are constitutive features of the faster developing, derived species (i.e., *S. couchii*)[13, 16, 17]. It appears that, since the last common ancestor of *S. couchii* and the other species, selection stabilized the short larval period phenotype that was previously obtained only by environmental induction.

At a mechanistic level, the rapid development of *S. couchii* may be explained by the higher tissue content of TH as well as increased sensitivity to TH action, due to higher expression of TH receptors[10, 33]. Also, reduced plasticity in larval period in *S. couchii* may be explained by the maintenance of high TH and CORT production at near climax levels throughout the larval period[9]. Thus, it is possible that selection to achieve short larval periods in ephemeral ponds via high hormone levels in *S. couchii* may have inadvertently resulted in reduced larval period plasticity as a consequence. A hypothetical alternative evolutionary pathway to achieve short larval periods is that *S. couchii* could have evolved a short larval period via evolution of rapid tissue transformation with reduced dependence on TH, such that none of the other changes associated with accelerated metamorphosis in plastic species, involving metabolism, limbs, gonads, and fat bodies, would necessarily have been obtained. Our data do not support this idea but rather provide rare endocrine evidence supporting the hypothesis that evolution of phenotypic differences among species occurred through genetic accommodation.

Variation in genome size among the species examined here and the existence of polyphenism in *Spea* such that its tadpoles may be induced to express an alternative carnivorous morphology had the potential to affect our current results. Genome size differences correlate with differences in larval period among species, such that species with faster development rate had reduced genome sizes[22]. To our knowledge, there is no known association between the level of plasticity and genome size across species, and this pattern is likely a by-product of selection to achieve faster development via more rapid cell division, also found in insects[34]. With regard to the expression of the induced carnivorous morphology, *S. multiplicata* can produce carnivorous tadpoles in response to appropriate environmental cues, including anostracan fairy shrimp and *Scaphiopus* tadpoles as food[35], but omnivore morphs seem to be produced as the default in nature[36], Even though carnivore morphs develop faster than omnivore morphs, survival through metamorphosis did not differ between carnivore and omnivore morphs in a pond drying experiment[37]. In any case, carnivore morphs were not observed in our experiments and thus did not impact the developmental rate or plasticity we observed in *Spea*. In addition, neither *Pelobates* nor *Scaphiopus* are known to produce carnivore morphs. On the other hand, the potential for *Spea* to produce carnivore morphs could have affected their evolutionary path differently from *Scaphiopus*, such that *Spea* would not "need" to develop rapidly, like *Scaphiopus*, because a faster developing carnivore morph is a developmental option[38].

In conclusion, our findings provide essential evidence for a link between the mechanisms controlling developmental plasticity and species phenotypic divergence. Because of their pivotal roles in controlling larval period and developmental plasticity, variation in TH and CORT production and signaling was a key factor in the evolution of larval period and plasticity in spadefoot toads. Our data provide mechanistic support for the hypothesis that selection for shorter larval periods with concomitant reduction in ancestral plasticity in *S. couchii* accounts for phenotypic divergence among spadefoot toad species.

## Methods

**Obtaining adult spadefoot toads and tadpoles.** Adults of *S. couchii* and *S. multiplicata* were collected from SE Arizona in the summer of 2010. Adult spadefoot toads were maintained in Cincinnati in screen-covered plastic boxes in 15 cm deep soil, fed ad libitum vitamin-dusted crickets, and sprinkled with water once a week. Two adult pairs from each species were taken to Seville, Spain for the experiments in March 2011. Adults were hormonally stimulated to breed by injecting intraperitoneally once with 20–100 μL of 1 μg/100 μL GnRH agonist (des-Gly[10],[D-His(Bzl)[6]]-luteinizing hormone releasing hormone ethylamide, Sigma)[13]. Two days after fertilization, tadpoles were transferred to large stock tanks with aeration and fed ad libitum finely powdered rabbit chow twice a day. *Pelobates cultripes* tadpoles (approx. Gosner stage 30[39]) were collected from Doñana National Park, Spain in March 2011. Tadpoles from all three species were reared at 25 °C with a 12L/12D cycle until they reached desired Gosner stages. The use of animals in these studies was approved by the University of Cincinnati Institutional Animal Care and Use Committee (IACUC protocol #06-10-03-01).

**Experimental set-up and design.** The species studied in this report live in very different natural habitats and thus it was very important to minimize the possibility that any given species was being favoured by the specific experimental setup when comparing effects of water reduction among species. We conducted the water reduction study starting at G35 and at 25 °C based on our previous study[17]. All tadpoles from a particular species and beginning stage were obtained and exposed to high and low water treatments on the same day. The high water treatment was 2.5 L of water per tank (130 mm water depth) for each species. The low water treatment in the same size tanks was determined by the volume of water required to just submerge the tadpole, and thus depths varied with species-specific tadpole size. Tadpoles were haphazardly assigned to treatments, reared individually, and fed ad libitum finely ground rabbit chow twice daily, and water was changed daily, as per previous studies which showed these rearing conditions were most favorable across species[13, 14, 17]. Each species is represented by at least three different clutches in each experiment, and specific sample sizes based on our previous studies with these species are provided in the figure legends. The researchers were blind as to the treatment of each specimen or sample at the time of conducting assays. Experiments were not repeated within the study. To study changes in SMR in response to water reduction as well as relation between CORT and SMR in tadpoles, we set up four treatments; tadpoles exposed to (1) high water+ETOH, (2) low water+ETOH, (3) high water+CORT (100 nM), (4) low water+metyrapone (25 μM). Corticosterone and metyrapone both were dissolved in ethanol and thus equal amount of ethanol was added to high and low water treatments to control for vehicle.

**Sample collection for hormone measurements.** To measure hormone content across species during development, whole tadpoles were collected at G35, 38 and 42 to measure CORT and at G38 and G42 to measure TH from large stock tanks (60 cm H×60 cm W×120 cm L) filled 3/4 full with water. To examine the endocrine response to water reduction treatment, whole tadpoles were collected after 24, 48, and 96 h after the beginning of water reduction treatment and at G38 and G42 (forelimb emergence, FLE) to measure CORT and TH. All tadpoles were snap frozen in liquid nitrogen after removing their intestine within 3 min of capture.

**Sample collection for metabolic rate determination.** To measure SMR during development, tadpoles from all three species were haphazardly chosen from large stock tanks (60 cm H×60 cm W×120 cm L) filled 3/4 full with water at G32, 35 38 and 42. To examine SMR response to water reduction and CORT manipulation, tadpoles were collected after 4 days to measure the rate of oxygen consumption. We estimated SMR as oxygen consumption at 25 °C using an aquatic respirometer system consisting of ten flow-through cells (plexiglass cylinders, 44 m in diameter×163 mm long cylinders) and twenty optical sensors connected to an oxymeter (Oxy 10-PreSens, Germany). Two sensors flanked each chamber to simultaneously measure oxygen concentration (mg/L) from the inflow and outflow of each chamber. A series of oxygen determinations were obtained every 15 s over the course of 45 min, averaged across the series. We systematically discarded the first 5 min of the data series, considered as acclimation period of the animals to the chambers. SMR values were calculated as $VO_2 = Vw \cdot \Delta Cw$, where $VO_2$ (μg h$^{-1}$) is the SMR measured as rate of oxygen consumption, Vw is the water flow rate through the chamber (1 h$^{-1}$), and $\Delta Cw$ is the instantaneous difference in $O_2$ concentration between the inflow and the outflow. All trials were conducted between 09.00 and 14.00 to avoid circadian effects. Upon release from the chambers, tadpoles were blotted dry and weighed to the nearest 0.1 mg on a high precision balance (CP324S, Sartorius, Germany). Individual body mass was used as a covariate in all statistical analyses of SMR[40, 41]. To measure CORT content, separate sets of tadpoles were collected at the same time and snap frozen in liquid nitrogen after removing their intestine within 3 min of capture.

**Phenotypic measurements.** Time in days from beginning of water reduction to FLE was measured to compare plasticity in the rate of metamorphosis across species.

**Hormone extraction and measurement.** CORT and thyroxine were extracted and measured using radioimmunoassay (RIA) as described previously[15, 25, 42, 43].

Estimates of whole body hormone content were corrected for recovery and sample volume, and are reported as pg/g*body weight. We euthanized tadpoles by immersion in 0.01% benzocaine, then placed them into 16 × 125 mm borosilicate tubes for snap freezing and storage at −80 °C before hormone extraction. Upon thawing, we removed the gastrointestinal (GI) tract (to eliminate variation in body weight determination due to differences among tadpoles in food and feces content of the GI tract), then recorded body weight for each tadpole. We extracted thyroxine ($T_4$) and CORT from the remaining carcass following methods described by Denver[15]. Briefly, we homogenized tadpoles in three to four volume of methanol containing 1 mM propylthiouracil (PTU; Sigma Chemical Co., St. Louis, MO). For *P. cultripes* and *S. multiplicata* tadpoles we extracted and measured hormone content in individual tadpoles; for *S. couchii* we pooled three tadpoles per sample owing to their small body size. For *P. cultripes* and *S. couchii* we divided the individual homogenates in two for measurement of $T_4$ or CORT. For *S. multiplicata* we only measured CORT.

For extraction of $T_4$, we first added 1000 cpm of [$^{125}$I]$T_4$ dissolved in 200 μl methanol (Perkin Elmer: formerly New England Nuclear; NEX111X) to monitor recoveries after extraction. Samples were incubated at room temperature for 15 min, then vortexed on a multitube vortexer for 10 min at room temperature followed by centrifugation at 1300 × *g* for 20 min at 4°C. The samples were then back-extracted by addition of 5 ml chloroform and 0.5 ml 2 N ammonium hydroxide, followed by vortexing for 15 min at room temperature. Samples were centrifuged at 1300 × *g* for 20 min at 4°C, then the aqueous phase was removed and applied to an ion-exchange chromatography column (Poly Prep; Bio-Rad, Hercules, CA) prepared with AG 1 × 2 resin (200–400 mesh; 1.5 ml bed volume; Bio-Rad) in 0.2 M acetate buffer (pH 7). The column was rinsed with alternating application of 0.2 M acetate buffer (pH 7) and 100% ethanol with 1 mM PTU, followed by elution with 70% acetic acid. The samples were dried in a SpeedVac concentrator, then resuspended in 0.5 ml radioimmunoassay buffer (0.11 M sodium barbital buffer, pH 8.6 containing 1 g/L bovine gamma-globulins, Cohn fraction II; Sigma). Fifty microliters of the resuspended sample were removed and counted in a gamma counter (Micromedic) to estimate recovery. We analyzed $T_4$ by RIA as described by MacKenzie et al.[44] All samples were analyzed in a single RIA with intra-assay variation less than 10%.

For extraction of CORT, we first added 3000 cpm of [1,2,6,7-$^3$H(N)]CORT (NET399) dissolved in 200 μl methanol to the homogenates to monitor recoveries after extraction. Samples were incubated at room temperature for 15 min, then 5 ml ethyl acetate was added, the samples were vortexed on a multitube vortexer for 10 min at room temperature, and centrifuged at 1300 × *g* for 20 min at 4°C. The supernatant was dried in a SpeedVac concentrator, and the pellet resuspended in 0.1 ml ethyl acetate for fractionation by thin-layer chromatography (TLC). Samples were loaded on silica gel GF TLC plates (Analtech 250 μM) and the plates placed in a mobile phase of toluene:cyclohexane (1:1). After development, the plates were air-dried and placed in a second mobile phase of chloroform:methanol (98:2). The plates were then air-dried and scanned for radioactivity with a Berthold TLC scanner to locate the [$^3$H]CORT peak. A 1 × 2-cm section of silica from each lane (corresponding to the [$^3$H]CORT peak) was scraped off of the plate into a 16 × 125 mm borosilicate tube, then 5 ml ether was added and the samples incubated overnight at 4 °C. The ether was dried under a stream of nitrogen and the sample re-dissolved in RIA buffer (0.2 M phosphate-buffered saline, pH 7 containing 1% gelatin). We analyzed CORT by RIA as described by Licht et al.[45] All samples were analyzed in a single RIA with intra-assay variation < 10%.

**Statistical analyses.** All analyses were conducted in R v3.2.1 (R Core Development Team). We used general linear models to test for differences across species, stages and/or experimental treatments on time to metamorphosis, hormone concentration (TH and CORT), and SMR. Species and developmental stage were included as fixed factors in the analyses, and their interaction tested (Supplementary Table 1). Within species, we conducted Fisher's LSD post hoc tests as needed, using the *glht* function from the library *multcomp*. Variables were log-transformed prior to analysis as needed to meet parametric assumptions. Data on time to metamorphosis and SMR showed substantial heteroscedasticity, which was corrected using robust standard errors with the function 'coeftest' in library *lmtest*.

**Data availability.** All relevant data are available from the corresponding authors upon request.

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

## Acknowledgements

We thank Arizona Game and Fish Commission (SP565423) and New Mexico Department of Game and Fish (3364) for issuing collecting permits for *Scaphiopus couchii*. We also thank Dr. David Pfennig for providing us with adult breeding pairs of *Spea multiplicata*. We also thank the Consejeria de Medio Ambiente from Junta de Andalucia for issuing collecting permits for *Pelobates cultripes*. Support for this research came from Dissertation Completion Fellowship from the University of Cincinnati and Journal of Experimental Biology Travel Fellowships to SSK (2009 and 2011) and NSF IOS 0950538 to D.R.B. and NSF IOS 0922583 to R.J.D. and grant CGL2014-59206-P from Plan Nacional I+D to I.G.M.

## Author contributions

I.G.-M. and D.R.B. collected animals, S.S.K. and I.G.-M. carried out tadpole rearing and treatments, metabolic rate measurements, and data analysis. S.S.K. and R.J.D. conducted and contributed to hormone measurements, S.S.K., I.G.-M., and D.R.B. designed the study and wrote the paper. All authors discussed the results and commented on the manuscript.

## Additional information

**Competing interests:** The authors declare no competing financial interests.

