## [Peer Review File · Nature Communications]

Reviewers' comments:

Reviewer #1 (Remarks to the Author):

In contrast to most studies of developmental plasticity, which focus on differences between populations, this study provides evidence for genetic accommodation contributing to species divergence. It shows that the loss of plasticity in larval period shown by one New World frog genus (*Scaphiopus*) relative to an Old World close relative (*Pelobates*) is associated with a lack of variability in the levels of hormones that regulate larval development and metabolism. In other words, in becoming adapted to a very short-lived larval habitat, *Scaphiopus* has apparently lost the ability to vary TH and corticosteroid levels in response to rate of pond drying, by becoming genetically fixed for high rates of hormone production, which result in constitutively high rates of larval development and metabolism.

The findings of this study should be of significant interest to amphibian ecologists, vertebrate endocrinologists studying the role of hormones in growth and development, and all biologists who are following the growing body of research on the role of phenotypic plasticity as a source of evolutionary change.

The paper is well written and I have only a few general and specific suggestions for revision.

First, line 30 Re "However, there is scarce evidence that such divergence can be translated above the species level, though some studies report patterns of species diversity that appear consistent with genetic accommodation".

What exactly is meant by "translated above the species level? I sense the terminology on this issue here and elsewhere in the paper (e.g., "genuine mechanism to bridge the gap" on line 163) is deliberately vague to avoid the implication that genetic accommodation is driving speciation. But perhaps the authors could be a bit clearer about what they mean. I agree that establishing a link between microevolutionary mechanisms and macroevolutionary differences is important, but can they tell us a little more about why it is important? Could genetic accommodation in this case have led to the emergence of reproduction isolation?

Second, Re Line 39: I had difficulty following the distinction being drawn between "species with ancestral plasticity in larval period" and "species with less responsive developmental plasticity". Do the authors mean species with high and low plasticity in larval period? Many readers might assume that the ancestral condition for amphibians is to have high plasticity, but this should be made clearer. More generally, the authors could explain their comparison and its interpretation a little more clearly in the introduction. It is not a difficult study to follow but the writing clouds it a little for the nonspecialist reader. For example, see my summary above.

Third, what are the stages for the data shown in Fig. 1C and D? This should be in the figure legend.

Fourth, re line 43-46: I don't understand the sentence. I think the authors in earlier papers have already tested the assumption that the phenotypic response triggered by decreasing water levels in *Scaphiopus* is matched by a shift in the underlying developmental endocrine mechanism. Why this is stated as an assumption of genetic accommodation is not clear to me. Also, the study does not address any genetic mechanisms, which are the assumed basis of genetic accommodation and which in this study could actually involve the perception of the environmental stress (water level) and not the neuroendocrine axis. In other words, the genetic accommodation could have happened at a pre-hypothalamus mechanism, and the endocrine axis could in fact still have some latent capacity for plasticity that is no longer being triggered by water level variation.

This point also leads to a larger issue, which is to really demonstrate that a developmental mechanism has been canalized to produce peak hormone levels right from the start of metamorphosis, doesn't one have to show that it is not possible to bring the hormone levels down at this stage, and that no other variables can affect the hormone levels. I know this opens up a Pandora's box of many mechanisms that are known to affect developmental rate in amphibians, and that are presumed to also act through the same neuroendocrine axis. For example, doesn't *Scaphiopus* express a carnivorous or cannibal morph (like *Spea*) and, if so, does this developmental switch correlate with speeded up development and metabolism as in *Spea*.

Also, just sticking to water level, what would happen if *Scaphiopus* tadpoles were raised in ponds much deeper than 13 cm – is it not possible that they might show lower TH and CORT levels at the start of metamorphosis than in the constant water depth (13 cm) treatment of this study, and longer larval periods. How deep are the ponds that *Pelobates* normally grow in?

Granted, the results are all consistent with genetic accommodation having occurred. Nonetheless, given that it is not possible to test for variation in response to all factors known to affect amphibian developmental rates, the authors should address the limits of their efforts to demonstrate canalization of the neuroendocrine activity and to pinpoint the exact developmental mechanisms at which genetic accommodation has apparently occurred.

Fifth, re line 148 and line 296 "metyrapone treatment failed to reduce CORT levels". What does this mean? That high levels of CORT produced before the metyrapone was applied persist during the treatment?

Chris Rose

Reviewer #2 (Remarks to the Author):

The major claim made in the paper by Kulkarni et al. is that patterns of development time and hormone levels in response to drought stress are correlated among species of spadefoot

toads. The species with the fastest developmental rate had the lowest plasticity of developmental time and hormone levels, whereas the species with the slowest developmental rate had the highest plasticity. The species from the ephemeral habitat that routinely experiences periods of drought showed no increase in corticosterone levels under drought stress, whereas the other two species did.

I do not feel that these results are novel enough to deserve publication in Nature Communications. It is already known that thyroid hormone and corticosterone influence developmental time, as cited in the paper. Thus, it is not surprising to find that they are linked to developmental time in this study. While comparisons among the species is interesting, an n of 3 species is too low to make generalizations about the relationship between habitat and plasticity (and this link has been found in several other studies, also cited in the paper).

Specific comments:

It is inappropriate to state that the "results establish a mechanistic link" with only three species, and without establishing hormonal variation is directly causing variation in developmental time.

Results would be easier to follow if they were put into a table. It is not clear to me how many individual ANOVAs were conducted, what factors were included in each ANOVA, and whether they were fixed or random factors. Importantly, there was no mention of testing for interactions among factors. This should be made more transparent.

Line 343: I think tadpoles would have been "haphazardly" chosen, not "randomly" chosen.

Reviewer #3 (Remarks to the Author):

This paper seeks to clarify the endocrine mechanisms that underlie genetic accommodation of developmental rate (and correlated traits) in spadefoot toads. As background, biologists have long recognized that phenotypic plasticity can produce new developmental variants that enhance fitness under stressful conditions. If underlying genetic variation exists in the tendency or manner in which individuals produce such variants, then selection can refine the trait from an initial, potentially suboptimal version through quantitative genetic changes over time, an evolutionary process known as genetic accommodation. Some have even argued that genetic accommodation can promote differences among species. However, there are actually few clear-cut examples of genetic accommodation in natural populations and fewer still known to account for phenotypic diversity among species.

Here, the authors compared hormonal profiles among three species of spadefoot toads that differ in degree of plasticity in larval developmental rate. They found that a species that shows pronounced plasticity in developmental rate exhibits increased variation in hormone levels. In contrast, the species that occur in more ephemeral environments and that show

less plasticity in larval developmental rate, constitutively produced high levels of these same hormones. The authors conclude that their data therefore “support for genetic accommodation beyond phenotypic comparisons by demonstrating that the mechanisms regulating developmental responses to environmental conditions in species with ancestral plasticity are canalized in descendent species.”

Generally, I enjoyed reading this paper and believe that it has the potential to make an important contribution to the literature on the underlying mechanisms of genetic accommodation. At the same time, I did have some serious reservations with the paper as it is presently written. These concerns revolve around: 1) how the authors set up the problem; 2) the degree to which the results are surprising; 3) what, if anything, is known about the direction of evolution in this system (i.e., decreased versus increased plasticity); and 4) more generally, whether they have actually demonstrated that developmental rate has undergone genetic accommodation in spadefoots. Below, I amplify on each of these points.

Major concerns:

1) First, I’m concerned with how the authors set up the problem. On lines 43-46, the authors state that “a ... key assumption of genetic accommodation ... [is that] ... a commonality of developmental endocrine mechanisms underlies both developmental plasticity and interspecific phenotypic differences.” Where does this assumption come from? Such a strong statement requires a reference, but none is provided. If it comes from West-Eberhard’s (2003) book, then we need page numbers.

I raise this issue, because, from my viewpoint, this is NOT a critical assumption of genetic accommodation. Genetic accommodation (as defined by West-Eberhard 2003, p. 140) is “gene frequency (evolutionary) change due to selection on variation in the regulation, form, or side effects of the novel trait in the subpopulation of individuals that express the trait.” Clearly, adaptive change in “the regulation, form, or side effects of a trait” could occur through either a single mechanism (as shown in this paper, as well as by, e.g., the study by Scoville and Pfrender on *Daphnia*) or, just as plausibly, through DIFFERENT mechanisms acting in ancestral versus derived lineages; i.e., it need NOT be the same mechanism in all lineages. In fact, at the genetic level, you can easily envision there being multiple mechanisms that produce the very same end result. It’s important to make this clear, because there’s already confusion over the key criteria of genetic accommodation, and I worry that inventing new criteria will unnecessarily muddle the field even further.

I think that there’s an easy fix for the authors here. In my view, what the authors are really doing in their study is exploring the underlying endocrine mechanisms of genetic accommodation in this system; such information is vital, because it helps identify the targets of selection that promote genetic accommodation. For example, the data presented here suggest that the genes and gene regulatory networks that mediate TH and CORT are targets of genetic accommodation of developmental rate in spadefoots. Very few studies have gotten this far, so this paper could provide a great contribution to the field on this basis alone.

2) Second, and related to the previous point above, the fact that (in the present case) the same endocrine mechanism underlies both developmental plasticity and interspecific phenotypic differences is not at all surprising. The authors state that they sought to establish whether (lines 44-46) "a commonality of developmental endocrine mechanisms underlies both developmental plasticity and interspecific phenotypic differences," but on lines 70-72, they state that "developmental acceleration in response to pond drying in spadefoot toads is, as in all other anurans, largely dependent upon increased levels of thyroid hormone (TH) and corticosterone (CORT)." Thus, these hormones were GUARANTEED to regulate developmental rate both within versus between species.

In other words, the underlying mechanisms MUST be the same within and between species. So, in a sense, the outcome was not surprising nor was it unanticipated. Again, this weakens the authors' assertion that a key assumption of genetic accommodation is that a commonality of developmental endocrine mechanisms must underlie both developmental plasticity and interspecific phenotypic differences – this will likely be true in MANY species at the broad level of endocrine mechanisms, where there is a limited number of hormones that could plausibly mediate such responses!

3) Third, what (if anything) is known about the directionality of evolution in this system? The authors state (e.g., in their first paragraph) that the direction of evolution is from an ancestral condition in which species were highly flexible in developmental rate in response to pond drying (as represented by modern-day *Pelobates*) to a derived condition in which species show little plasticity in developmental rate (as represented by modern-day *Scaphiopus couchii*). What is the evidence that modern-day *Pelobates* does indeed represent the ancestral condition and that modern-day *Scaphiopus couchii* represents a derived condition? Certainly the phylogeny in Figure 1A does not tell us this. Also, it cannot be claimed that *Spea multiplicata* is ancestral to *Scaphiopus couchii* (as the authors imply), since these two species are from sister genera.

4) Finally, based on this study and the authors' previous work, can we actually say with certainty that developmental rate per se has undergone genetic accommodation in spadefoots? As the authors note (line 29), the process of genetic accommodation is driven by SELECTION. However, I'm not aware of any study that has actually measured selection on developmental rate per se in spadefoots and has shown, for example, that modern-day *Scaphiopus couchii* have experienced directional selection for faster developmental rate (and hence, less plastic timing of metamorphosis / hormone profiles). It makes sense that this MIGHT be true, given the environments that *Scaphiopus couchii* inhabits (but see comment 6 below).

However, without direct measurements of selection in the wild (or in controlled lab environments), one could not rule out the alternative hypothesis that divergence among species of spadefoots in developmental rate (and degree of plasticity in this rate) has evolved through a process of "species (or "lineage") sorting". Specifically, species (or lineages) that were already fast developers were more likely to invade habitats with ephemeral ponds than those that were slow developers. Such species sorting can arise

either through the differential invasion into ephemeral habitats by fast developers or through the differential extinction of slow developers. Regardless of how it came about, species sorting is not a mechanism of evolution (i.e., it is not a selective process), because it does not entail trait evolution per se. Demonstrating that SELECTION – acting on variation within species – seems to be missing here.

More generally, demonstrating that selection has promoted an evolutionary change in reaction norms is missing in many studies of genetic accommodation (e.g., most of the studies cited in Schlichting and Wund's review of genetic accommodation actually lack such information).

Additional concerns:

5) As the authors are aware, *Spea multiplicata* produces alternative larval ecomorphs (as part of a polyphenism) that also differ in developmental rate: a slow-developing omnivore morph and a more rapidly-developing carnivore morph. Could the existence of this omnivore-carnivore polyphenism explain why much greater levels of plasticity were observed in *Spea* than in *Scaphiopus* (because the former included both morphs)? If the data presented in this paper are from omnivores only, how does the exclusion of carnivores affect the results, since most natural populations would actually produce both morphs? It seems that excluding carnivores would give one a greatly biased representation of developmental rate (and the correlated traits) in *Spea*.

6) Contrary to what the authors claim here, *Spea multiplicata* and *Scaphiopus couchii* do NOT appear to be experiencing "divergent environments", at least not divergent hydroperiod environments, which is implied by the authors. The two species have nearly completely overlapping geographical distributions (in the U.S. at least), and, indeed, their tadpoles often co-occur in the SAME ponds. Why, then, have they diverged so much in developmental rate (and associated traits)? This seems to be a major gap in the authors' story on this system. Some other agent(s) of selection must be in play here (this point is also related to my comment 4 above).

7) The data on standard metabolic rate (SMR) were interesting, but it was not clear how these data (lines 117-119) "further support the hypothesis that the endocrine mechanisms underlying the plastic acceleration of metamorphosis in *P. cultripes* and the canalized rapid development in *S. couchii* are equivalent." We need more explanation of why these sort of traits were included in the analyses.

8) For the reasons outlined above, I'm not sure I agree with the authors' claim that their "current physiological results and our previous morphological results combine to provide the strongest evidence to date showing how genetic accommodation affecting developmental plasticity may have given rise to evolutionary differences among species in divergent environments." However, whether one agrees with this statement or not, it is always best to let data speak for itself. Thus, I would remove such statements from the paper.

Minor concerns and suggestions.

On line 33, the authors state that they “show support for genetic accommodation beyond phenotypic comparisons”. However, this study shows “phenotypic comparisons”; it’s just at a different level of phenotype that the authors’ earlier studies on this system.

Line 38: Should say, “spadefoot toad species”

Line 66: I would be careful about saying that *Scaphiopus* has “little or no fat”; what you’re really talking about here are abdominal fat bodies. In addition to these fat bodies, frogs (including spadefoots) store fat in their liver as well as subcutaneously.

Line 69: CORT should be defined here

Line 163: What do you mean by the phrase “genuine mechanism”? I would remove the word “genuine” as this may be construed by some to suggest that the mechanisms that other researchers have identified are not “genuine.”

Figures generally: The figures were hard for me to read in the 8 x 10-inch format at which I printed them out, and the authors should be aware of this. Also, I’m not sure that the box plots need to be shown in different colors to represent the different species – doing so just adds to the overall “noise” in the figures.

Figure 1 (B,C,D): It would be useful to see the actual values somewhere. Generally, for analyses, I’m curious as to why the authors did not perform a multiple comparisons test across all groups (in addition to, or instead of, doing pairwise tests within species). This seems like an unorthodox approach to me. I’m also concerned about the amount of variation in the “canalized” species in D. Of course, if there’s a lot of variation, you would not expect to find a significant difference in mean. Probably not a death stroke, but I think it’s important to recognize this when discussing the paper’s significance.

Figure 2: It doesn’t say what conditions these data were taken from: high or low water?

Figure 3: I would like more explanation about the metyrapone treatment. If it is supposed to prevent synthesis of CORT, then why wouldn’t it do that in *S. couchii*?

Line 310: Since the *P. cultripes* tadpoles were wild caught, how can the authors be certain that the developmental rate “decision” of their *P. cultripes* tadpoles wasn’t influenced by their early life experience/environment?

Line 327: “The sample size varied between 8 and 10 for each species.” Are these families, tadpoles? Multiple populations? The answer to this question can be important because the degree of plasticity can vary at all these levels.

Line 330-331: What was the volume of the 100nM CORT and 25nM metyrapone used?

Line 336-337,343-344: "large stock tanks" What is this referring to?

Lines 368-371: Was the mass of each sample at least the same? Wouldn't it have been better to do some size-adjusted measurement? Or the same volume of homogenate?

The figures look like it is picograms of hormone per milligram of tissue. So is it higher amounts of hormone or lower amounts of tissue driving the pattern?

Reviewers' comments:

Reviewer #1 (Remarks to the Author):

In contrast to most studies of developmental plasticity, which focus on differences between populations, this study provides evidence for genetic accommodation contributing to species divergence. It shows that the loss of plasticity in larval period shown by one New World frog genus (*Scaphiopus*) relative to an Old World close relative (*Pelobates*) is associated with a lack of variability in the levels of hormones that regulate larval development and metabolism. In other words, in becoming adapted to a very short-lived larval habitat, *Scaphiopus* has apparently lost the ability to vary TH and corticosteroid levels in response to rate of pond drying, by becoming genetically fixed for high rates of hormone production, which result in constitutively high rates of larval development and metabolism.

The findings of this study should be of significant interest to amphibian ecologists, vertebrate endocrinologists studying the role of hormones in growth and development, and all biologists who are following the growing body of research on the role of phenotypic plasticity as a source of evolutionary change.

The paper is well written and I have only a few general and specific suggestions for revision.

Response: We thank the reviewer for the careful and constructive analysis of the manuscript. Below are our best efforts to address the specific concerns.

First, line 30 Re "However, there is scarce evidence that such divergence can be translated above the species level, though some studies report patterns of species diversity that appear consistent with genetic accommodation".

What exactly is meant by "translated above the species level? I sense the terminology on this issue here and elsewhere in the paper (e.g., "genuine mechanism to bridge the gap" on line 163) is deliberately vague to avoid the implication that genetic accommodation is driving speciation. But perhaps the authors could be a bit clearer about what they mean. I agree that establishing a link between microevolutionary mechanisms and macroevolutionary differences is important, but can they tell us a little more about why it is important? Could genetic accommodation in this case have led to the emergence of reproduction isolation?

Response: We appreciate the comment and the opportunity to clarify these sentences. We have no knowledge of and did not intend to make a claim about a possible relationship between selection for short larval periods and reproductive isolation. In general, speciation and phenotypic divergence among species are not necessarily correlated or mechanistically (developmentally) linked, though they could be. We are trying to explain the evolutionary/developmental mechanism to explain why species may differ in phenotype, but not why they are different species. Following your suggestion we have rewritten the sentence as (line 55-58):

"However, how mechanisms of trait regulation evolve during genetic accommodation, as we show here for spadefoot toads, is much less understood and of vital importance for elucidating why/how lineages differ in phenotype."

Also, at the end of the discussion (line 197-200): "Our data add mechanistic support to the view that selection has resulted in canalization of ancestral plasticity and caused phenotypic divergence among species. This may be a common mechanism linking trait evolution at both micro- and macroevolutionary scales".

Second, Re Line 39: I had difficulty following the distinction being drawn between “species with ancestral plasticity in larval period” and “species with less responsive developmental plasticity”. Do the authors mean species with high and low plasticity in larval period? Many readers might assume that the ancestral condition for amphibians is to have high plasticity, but this should be made clearer. More generally, the authors could explain their comparison and its interpretation a little more clearly in the introduction. It is not a difficult study to follow but the writing clouds it a little for the nonspecialist reader. For example, see my summary above.

Response: Yes, we mean just high and low plasticity in the larval period. That section now reads: "We found that ancestral mechanisms regulating developmental responses to environmental conditions characterized by a high level of larval period plasticity evolved in descendant species, giving rise to a canalized rapid developmental rate". (line 58-61) We made extensive changes to the introductory and conclusion paragraphs to improve the presentation of the question and results of the manuscript to accommodate reviewer suggestions.

Third, what are the stages for the data shown in Fig. 1C and D? This should be in the figure legend.

Response: We now state on the figure legend that hormonal assays were conducted on tadpoles at Gosner stage 38.

Fourth, re line 43-46: I don't understand the sentence. I think the authors in earlier papers have already tested the assumption that the phenotypic response triggered by decreasing water levels in *Scaphiopus* is matched by a shift in the underlying developmental endocrine mechanism. Why this is stated as an assumption of genetic accommodation is not clear to me. Also, the study does not address any genetic mechanisms, which are the assumed basis of genetic accommodation and which in this study could actually involve the perception of the environmental stress (water level) and not the neuroendocrine axis. In other words, the genetic accommodation could have happened at a pre-hypothalamus mechanism, and the endocrine axis could in fact still have some latent capacity for plasticity that is no longer being triggered by water level variation.

Response: In revising the introduction, we removed the idea about a shared mechanism being a required assumption in genetic accommodation. Instead, we focus on establishing the underlying endocrine mechanisms of genetic accommodation in this system. We differ with the reviewer in thinking this has been done before. Previous work by us showed 1) hormone measurements (TH only) in all species in high water, 2) phenotypic comparisons among spadefoot species in high and low water, 3) hormone measurements in high and low water only in *Pelobates* by us and only in *Spea* by others, but not in *Scaphiopus*. To compare among species, we performed hormone measurements among all species in high and low water reared in the same laboratory conditions using a single assay protocol.

The reviewer is correct in pointing out that we have not shown the genetic basis of difference in plasticity among species, but we have shown distinct, species-specific developmental endocrine responses (or not) to water level. The reviewer is also correct that canalization in *Scaphiopus* could have been due to some sort of loss of a sensory response to pond drying, but in that case there would be no reason to expect higher constitutive levels of hormones in the canalized species, as we observe here. However, it is possible, but less parsimonious, that *Scaphiopus* ancestors had mechanistically independent genetic changes to 1) lose their stress response to pond drying (a lack of sensory processing of water level) and 2) gain (at the same time or a later time) constitutive high stress hormone levels (in a way not due to genetic fixation of high stress-induced levels of hormones). However, population genetic theory suggests that such multiple, independent genetic changes are less likely compared to a scenario of genetic accommodation. We observed the pattern that the same mechanism is involved in within-species plastic responses to environmental changes and among-species trait variation. Thus, as we state in the manuscript (line 191-185): "The interspecific

differences we found in endocrine regulation of development rate and larval period plasticity ... lend strong support that genetic accommodation on ancestral developmental plasticity has given rise to evolutionary differences among species in divergent environments."

This point also leads to a larger issue, which is to really demonstrate that a developmental mechanism has been canalized to produce peak hormone levels right from the start of metamorphosis, doesn't one have to show that it is not possible to bring the hormone levels down at this stage, and that no other variables can affect the hormone levels. I know this opens up a Pandora's box of many mechanisms that are known to affect developmental rate in amphibians, and that are presumed to also act through the same neuroendocrine axis. For example, doesn't *Scaphiopus* express a carnivorous or cannibal morph (like *Spea*) and, if so, does this developmental switch correlate with speeded up development and metabolism as in *Spea*.

Also, just sticking to water level, what would happen if *Scaphiopus* tadpoles were raised in ponds much deeper than 13 cm – is it not possible that they might show lower TH and CORT levels at the start of metamorphosis than in the constant water depth (13 cm) treatment of this study, and longer larval periods. How deep are the ponds that *Pelobates* normally grow in?

Response: Cannibal morphs are not known in *Scaphiopus*. To the larger questions, we have not done the experiment suggested (directly compare hormone content in *Scaphiopus* reared in 13 cm vs deeper water levels). In our experiments, tadpoles were kept individually in 3 L buckets and the difference between 'High' and 'Low' water volumes was enough to trigger acceleration in *Pelobates* and *Spea*, which are much larger tadpoles and typically live in much larger ponds. If the 'High' water level had been perceived by these species as if it were a 'Low' treatment, we would not have observed developmental acceleration. Natural history observations by others show that *Scaphiopus* tadpoles metamorphose quickly in natural ponds (typically 10-50 cm deep but up to 1 m deep) in less than two weeks even if pond duration is much longer and habitat quality is favorable (*Oikos* 104:172, *Oecologia* 71:301). Also, we and others have reared *Scaphiopus* and *Spea* in the lab varying temperature, food type, tadpole density, and food level, and *Scaphiopus* maintained its relatively invariant larval period duration. Hormone measurements haven't been made in all these conditions, but we suggest that minimal change in larval period duration is likely accompanied by minimal change in hormone levels as we have shown here at least for high and low water. Furthermore, we used tadpoles reared in large stock tanks about 50-60 cm deep for our comparisons of hormone levels among species across stages, where premetamorphic *Scaphiopus* CORT levels were higher than the other taxa as was found in the experimental buckets.

Granted, the results are all consistent with genetic accommodation having occurred. Nonetheless, given that it is not possible to test for variation in response to all factors known to affect amphibian developmental rates, the authors should address the limits of their efforts to demonstrate canalization of the neuroendocrine activity and to pinpoint the exact developmental mechanisms at which genetic accommodation has apparently occurred.

Response: We completely agree with the reviewer. Our manuscript adds developmental endocrine mechanisms to the previous morphological support for genetic accommodation in spadefoot toads, but other possibilities have not been completely ruled out as stated in our response above. Additional studies such as genetic/genomic support showing changes in the neuroendocrine regulation of development among species consistent with genetic accommodation would further strengthen the hypothesis of genetic accommodation. We now state (line 194-197): "Our results are critical for guiding genomic analyses to identify genetic and regulatory aspects of traits that may be the target of selection under divergent environmental conditions in order to expand our mechanistic understanding of genetic accommodation in this system."

Fifth, re line 148 and line 296 “metyrapone treatment failed to reduce CORT levels”. What does this mean? That high levels of CORT produced before the metyrapone was applied persist during the treatment?

Response: In *Pelobates* and *Spea*, the increase in CORT content in response to reduced water levels was blocked by metyrapone. However, *S. couchii* did not increase its CORT synthesis leading to unchanged CORT content even after applying metyrapone. We have clarified this in the text (lines 171-172).

Reviewer #2 (Remarks to the Author):

The major claim made in the paper by Kulkarni et al. is that patterns of development time and hormone levels in response to drought stress are correlated among species of spadefoot toads. The species with the fastest developmental rate had the lowest plasticity of developmental time and hormone levels, whereas the species with the slowest developmental rate had the highest plasticity. The species from the ephemeral habitat that routinely experiences periods of drought showed no increase in corticosterone levels under drought stress, whereas the other two species did.

I do not feel that these results are novel enough to deserve publication in Nature Communications. It is already known that thyroid hormone and corticosterone influence developmental time, as cited in the paper. Thus, it is not surprising to find that they are linked to developmental time in this study. While comparisons among the species is interesting, an n of 3 species is too low to make generalizations about the relationship between habitat and plasticity (and this link has been found in several other studies, also cited in the paper).

Response: We think that our study has broader implications than just showing that CORT and TH influence developmental timing. The question is *how/why* did the three species come to differ in the way that they do, and specifically, what is the explanation for why *Scaphiopus* has high constitutive CORT *and* low plasticity? More generally, no previous study has demonstrated developmental mechanisms supporting genetic accommodation in a natural setting to explain species phenotypic differences. Here, we show that species differ in their responsiveness to pond drying and that the underlying endocrine mechanisms vary among species exactly in the way we would predict if the plastic response common to most anurans had been canalized under directional selection for fast development. CORT, TH, and metabolic rate increase in plastic species in response to reduced water level, but all three traits are at constitutively high levels in the canalized species, precisely as one would expect after genetic accommodation.

Specific comments:

It is inappropriate to state that the “results establish a mechanistic link” with only three species, and without establishing hormonal variation is directly causing variation in developmental time.

Response: We have toned down this statement to: "The identification of endocrine mechanisms underlying genetic accommodation of development among spadefoot toad species provides a basis for understanding mechanisms that may link microevolutionary responses to local environments within species and macroevolutionary patterns of trait divergence among species. (line 34-37)" For our purpose, i.e., to look for a link between underlying developmental mechanisms and evolutionary divergence in larval period and plasticity supporting genetic accommodation, these three species, which include a highly responsive species, a canalized one, and an intermediate one, seem sufficient.

Our statement was not meant to provide a broad generalization about a relationship between larval period duration and hormone levels. It is possible that high TH content has no role in reduced larval period in *Scaphiopus*, because we didn't carry out such experiments, which are challenging to do conclusively. However, based on well-established influence of TH and CORT on larval period duration within species, it seems that hormone level variation did contribute to interspecific larval period differences.

Results would be easier to follow if they were put into a table. It is not clear to me how many individual ANOVAs were conducted, what factors were included in each ANOVA, and whether they were fixed or random factors. Importantly, there was no mention of testing for interactions among factors. This should be made more transparent.

Response: In this revised version we have added details to the description of the statistical analyses and a table showing results of overall model testing. For each of the different aspects of this study we conducted independent experiments in each species. For instance, we conducted an experiment to determine tissue content of hormones at different larval stages for each species, and each species was not necessarily tested simultaneously. Then we conducted a different experiment for each species to test the effect of reducing water volume on their hormonal levels. Yet another experiment tested for the effect of reduced water level on standard metabolic rate and corticosterone and metyrapone treatments. Therefore, because each was a different experiment with its own internal experimental controls, we conducted a number of general linear models, but they were all applied to independent datasets and therefore no adjustments for multiple testing was needed. In all analyses we used species, developmental stage, or experimental treatment as fixed factors. Since experiments were independent for each species, we tested for differences across stages or experimental treatments within species. For completion we have now conducted overall models testing for interactions between species and either stages or treatments, now included as a table in supporting information.

Line 343: I think tadpoles would have been “haphazardly” chosen, not “randomly” chosen.

Response: We corrected the text to "haphazardly".

Reviewer #3 (Remarks to the Author):

This paper seeks to clarify the endocrine mechanisms that underlie genetic accommodation of developmental rate (and correlated traits) in spadefoot toads. As background, biologists have long recognized that phenotypic plasticity can produce new developmental variants that enhance fitness under stressful conditions. If underlying genetic variation exists in the tendency or manner in which individuals produce such variants, then selection can refine the trait from an initial, potentially suboptimal version through quantitative genetic changes over time, an evolutionary process known as genetic accommodation. Some have even argued that genetic accommodation can promote differences among species. However, there are actually few clear-cut examples of genetic accommodation in natural populations and fewer still known to account for phenotypic diversity among species.

Here, the authors compared hormonal profiles among three species of spadefoot toads that differ in degree of plasticity in larval developmental rate. They found that a species that shows pronounced plasticity in developmental rate exhibits increased variation in hormone levels. In contrast, the species that occur in more ephemeral environments and that show less plasticity in larval developmental rate, constitutively produced high levels of these same hormones. The authors conclude that their data therefore “support for genetic accommodation beyond phenotypic comparisons by demonstrating that

the mechanisms regulating developmental responses to environmental conditions in species with ancestral plasticity are canalized in descendent species.”

Generally, I enjoyed reading this paper and believe that it has the potential to make an important contribution to the literature on the underlying mechanisms of genetic accommodation. At the same time, I did have some serious reservations with the paper as it is presently written. These concerns revolve around: 1) how the authors set up the problem; 2) the degree to which the results are surprising; 3) what, if anything, is known about the direction of evolution in this system (i.e., decreased versus increased plasticity); and 4) more generally, whether they have actually demonstrated that developmental rate has undergone genetic accommodation in spadefoots. Below, I amplify on each of these points.

Response: We thank the reviewer for the thorough analysis of the manuscript and encouraging comments regarding the potential of the study to make an important contribution. We make our best effort to address the issues raised on a point-by-point basis below.

Major concerns:

1) First, I'm concerned with how the authors set up the problem. On lines 43-46, the authors state that “a ... key assumption of genetic accommodation ... [is that] ... a commonality of developmental endocrine mechanisms underlies both developmental plasticity and interspecific phenotypic differences.” Where does this assumption come from? Such a strong statement requires a reference, but none is provided. If it comes from West-Eberhard's (2003) book, then we need page numbers.

I raise this issue, because, from my viewpoint, this is NOT a critical assumption of genetic accommodation. Genetic accommodation (as defined by West-Eberhard 2003, p. 140) is “gene frequency (evolutionary) change due to selection on variation in the regulation, form, or side effects of the novel trait in the subpopulation of individuals that express the trait.” Clearly, adaptive change in “the regulation, form, or side effects of a trait” could occur through either a single mechanism (as shown in this paper, as well as by, e.g., the study by Scoville and Pfrender on *Daphnia*) or, just as plausibly, through DIFFERENT mechanisms acting in ancestral versus derived lineages; i.e., it need NOT be the same mechanism in all lineages. In fact, at the genetic level, you can easily envision there being multiple mechanisms that produce the very same end result. It's important to make this clear, because there's already confusion over the key criteria of genetic accommodation, and I worry that inventing new criteria will unnecessarily muddle the field even further.

I think that there's an easy fix for the authors here. In my view, what the authors are really doing in their study is exploring the underlying endocrine mechanisms of genetic accommodation in this system; such information is vital, because it helps identify the targets of selection that promote genetic accommodation. For example, the data presented here suggest that the genes and gene regulatory networks that mediate TH and CORT are targets of genetic accommodation of developmental rate in spadefoots. Very few studies have gotten this far, so this paper could provide a great contribution to the field on this basis alone.

Response: For some time now we have been interested in the problem of how to recognize when trait evolution has occurred through genetic accommodation given some observed trait divergence among species. Our expectation that the same mechanisms regulating alternative developmental pathways within populations in response to environmental conditions should be responsible for among-lineage divergence largely stems from M.J. West-Eberhard's discussion on developmental recombination and the origin of species differences (West-Eberhard 2005, PNAS 102: 6543-6549). In her own words (op. cit), '*Phenotypic differences that eventually distinguish species may often arise before the advent of reproductive isolation between them, because the origin and maintenance of*

more than one developmental pathway can occur within a population; the evolution of a divergent novelty does not require gene-pool divergence, only developmental-pathway and gene-expression divergence (West-Eberhard 2003)'; and also 'Research on patterns of gene expression makes it possible to pinpoint the (expressed) loci that are actually subject to selection in the evolution of species differences, beginning with differences that arise because of developmental recombination without reproductive isolation.'

We have given quite some thought to the point raised by the reviewer, and although we still think that adaptive evolution through accommodation of developmental plasticity necessarily starts within population as modifications in the regulation of a given developmental pathway (or sets of pathways) and hence, at least initially, the phenotypic divergence among different lineages has to do with alternative regulations of the same developmental mechanism(s). However, we agree with the reviewer that evolutionary divergence of that developmental mechanism, i.e. genetic accommodation, could occur through different genetic modifications in different lineages since multiple combinations of genetic modifications could potentially cause changes in the regulation of the evolving trait. We have hence removed the statement regarding the shared mechanism as a key assumption in genetic accommodation. We maintain, however, that trait divergence among species is likely to be initiated as alternative phenotypes within species induced by developmental alterations and stress the need to identify the developmental pathways undergoing genetic accommodation. We thank the reviewer for suggesting this change in focus.

2) Second, and related to the previous point above, the fact that (in the present case) the same endocrine mechanism underlies both developmental plasticity and interspecific phenotypic differences is not at all surprising. The authors state that they sought to establish whether (lines 44-46) "a commonality of developmental endocrine mechanisms underlies both developmental plasticity and interspecific phenotypic differences," but on lines 70-72, they state that "developmental acceleration in response to pond drying in spadefoot toads is, as in all other anurans, largely dependent upon increased levels of thyroid hormone (TH) and corticosterone (CORT)." Thus, these hormones were GUARANTEED to regulate developmental rate both within versus between species.

In other words, the underlying mechanisms MUST be the same within and between species. So, in a sense, the outcome was not surprising nor was it unanticipated. Again, this weakens the authors' assertion that a key assumption of genetic accommodation is that a commonality of developmental endocrine mechanisms must underlie both developmental plasticity and interspecific phenotypic differences – this will likely be true in MANY species at the broad level of endocrine mechanisms, where there is a limited number of hormones that could plausibly mediate such responses!

Response: The reviewer is correct that CORT and TH were in a way guaranteed to have played a role in the evolution of developmental rate in this group. This is a strength of this system and not a weakness, because previous studies have laid the groundwork to study the evolution of the endocrine system in a phylogenetic perspective. However, because endocrine regulation is very complex, the molecular/endocrine/developmental changes possible to give rise to a species with tadpoles that can survive in ephemeral pools are numerous and not necessarily the species differences we found. The key issue to address is how the endocrine regulation changed, beyond the fact that CORT and TH were involved. Many possible endocrine changes are imaginable, but the changes we found precisely matched expectations of genetic accommodation where the hormone state observed in the plastic species undergoing accelerated development in drying ponds was commensurate with the hormone state of the species with constitutively rapid development. This situation is clearly consistent with genetic accommodation and point to potential targets of selection for further investigation to provide additional support for genetic accommodation and explain the evolution of trait diversification among species.

3) Third, what (if anything) is known about the directionality of evolution in this system? The authors state (e.g., in their first paragraph) that the direction of evolution is from an ancestral condition in which species were highly flexible in developmental rate in response to pond drying (as represented by modern-day *Pelobates*) to a derived condition in which species show little plasticity in developmental rate (as represented by modern-day *Scaphiopus couchii*). What is the evidence that modern-day *Pelobates* does indeed represent the ancestral condition and that modern-day *Scaphiopus couchii* represents a derived condition? Certainly the phylogeny in Figure 1A does not tell us this. Also, it cannot be claimed that *Spea multiplicata* is ancestral to *Scaphiopus couchii* (as the authors imply), since these two species are from sister genera.

Response: In an earlier paper (Gomez-Mestre and Buchholz PNAS 2006) we conducted an ancestral trait reconstruction and found that fast development in *Spea* and *Scaphiopus* was indeed derived and that the reaction norm of *Pelobates* closely matched the inferred ancestral reaction norm for all spadefoot toads. This notion is also supported by the fact that developmental plasticity in response to pond drying is actually fairly common in anurans, and that *Scaphiopus* (especially *S. couchii*) show the fastest development known within the whole Anura class, evidencing its departure from the norm. Also, we don't want to say *Spea* is ancestral to *Scaphiopus*, because as the reviewer indicates, they are species of sister genera. We merely note that *Spea* does show levels of developmental plasticity intermediate between the ancestral state and the even more extreme *Scaphiopus*. This is now referenced in line 74.

4) Finally, based on this study and the authors' previous work, can we actually say with certainty that developmental rate per se has undergone genetic accommodation in spadefoots? As the authors note (line 29), the process of genetic accommodation is driven by SELECTION. However, I'm not aware of any study that has actually measured selection on developmental rate per se in spadefoots and has shown, for example, that modern-day *Scaphiopus couchii* have experienced directional selection for faster developmental rate (and hence, less plastic timing of metamorphosis / hormone profiles). It makes sense that this MIGHT be true, given the environments that *Scaphiopus couchii* inhabits (but see comment 6 below).

However, without direct measurements of selection in the wild (or in controlled lab environments), one could not rule out the alternative hypothesis that divergence among species of spadefoots in developmental rate (and degree of plasticity in this rate) has evolved through a process of "species (or "lineage") sorting". Specifically, species (or lineages) that were already fast developers were more likely to invade habitats with ephemeral ponds than those that were slow developers. Such species sorting can arise either through the differential invasion into ephemeral habitats by fast developers or through the differential extinction of slow developers. Regardless of how it came about, species sorting is not a mechanism of evolution (i.e., it is not a selective process), because it does not entail trait evolution per se. Demonstrating that SELECTION – acting on variation within species – seems to be missing here.

More generally, demonstrating that selection has promoted an evolutionary change in reaction norms is missing in many studies of genetic accommodation (e.g., most of the studies cited in Schlichting and Wund's review of genetic accommodation actually lack such information).

Response: The reviewer is right in that we do not explicitly test for selection in this study, but instead we assume it because of the following argument. Some shifts in developmental rate within species are passive as in those resulting from changes in temperature, whereas others involve the activation of specific neuroendocrine pathways such as the HPA-axis that result in true adaptive responses to an environmental stimulus. This latter is the case when tadpoles respond to risk of desiccation from pond drying. Adaptive divergence in developmental rates among amphibian populations exposed to

different hydroperiods (i.e. selection for divergent developmental rates) has been demonstrated in island populations of *Rana temporaria* (Lind & Johansson 2007 J Evol Biol 20(4), 1288-1297). Ongoing work at the Gomez-Mestre Lab is testing for such differences too among spadefoot toad populations, but although we have indications of population differentiation in both their overall developmental rate and their response to reduced water volume, it is still work in progress. We have incorporated a sentence referencing Lind and Johansson stating that "developmental rate can diverge under selection in response to pond duration" (line 83). Also, we stress that *S. couchii* has evolved an extreme case of rapid development (it is the fastest known in anurans) typically associated with ephemeral ponds.

Additional concerns:

5) As the authors are aware, *Spea multiplicata* produces alternative larval ecomorphs (as part of a polyphenism) that also differ in developmental rate: a slow-developing omnivore morph and a more rapidly-developing carnivore morph. Could the existence of this omnivore-carnivore polyphenism explain why much greater levels of plasticity were observed in *Spea* than in *Scaphiopus* (because the former included both morphs)? If the data presented in this paper are from omnivores only, how does the exclusion of carnivores affect the results, since most natural populations would actually produce both morphs? It seems that excluding carnivores would give one a greatly biased representation of developmental rate (and the correlated traits) in *Spea*.

Response: This is a very interesting topic. Neither *Pelobates* nor *Scaphiopus* are known to produce carnivore morphs, and carnivore morphs are exceedingly rare in *Spea* under laboratory rearing conditions (<1 per 1000, even when trying to produce them by altering tadpole density and using anostocan shrimp, personal obs.). Even in nature, *Spea multiplicata* seem to produce omnivore morphs as default unless environmental cues, such as shrimp, are present. Thus, because our experiments depended on identical laboratory conditions, carnivore morphs were not observed and thus not actively excluded and therefore do not explain the greater levels of plasticity we observed in *Spea*. It would be fascinating to compare the developmental rate of both morphs of *S. multiplicata* exposed to reduced water level, but this has not been done to our knowledge. The option for *Spea* to produce carnivores (which are indeed reported to develop faster) may have affected their evolutionary path differently than *Scaphiopus*, such that *Spea* doesn't "need" to develop constitutively quickly, like *Scaphiopus* does, because a faster developing morph is a developmental option. However, pond drying does not seem to induce the carnivore morph, except perhaps indirectly via increasing the density of shrimp. These issues regarding *Spea*, however, leave our conclusions regarding *Pelobates* and *Scaphiopus* intact and may not necessarily impact the results or conclusions we obtained for *Spea*. In any case, this is an intriguing area to explore further.

6) Contrary to what the authors claim here, *Spea multiplicata* and *Scaphiopus couchii* do NOT appear to be experiencing "divergent environments", at least not divergent hydroperiod environments, which is implied by the authors. The two species have nearly completely overlapping geographical distributions (in the U.S. at least), and, indeed, their tadpoles often co-occur in the SAME ponds. Why, then, have they diverged so much in developmental rate (and associated traits)? This seems to be a major gap in the authors' story on this system. Some other agent(s) of selection must be in play here (this point is also related to my comment 4 above).

Response: It is true that there is extensive geographical and pond overlap between *Spea* and *Scaphiopus*, but in addition *Scaphiopus* will lay eggs in virtually any pond after heavy rains and does not seem to distinguish between ponds that will dry in 3 days versus 2 weeks or longer. *Spea* is more selective and does not lay eggs in ponds as ephemeral as *Scaphiopus* does. Thus, these species do differ in where their eggs end up (with some overlap), such that the tadpoles in the two species do

experience on average different hydroperiods.

7) The data on standard metabolic rate (SMR) were interesting, but it was not clear how these data (lines 117-119) “further support the hypothesis that the endocrine mechanisms underlying the plastic acceleration of metamorphosis in *P. cultripes* and the canalized rapid development in *S. couchii* are equivalent.” We need more explanation of why these sort of traits were included in the analyses.

Response: SMR is influenced by exogenous CORT in amphibians. Also, two of our previous studies suggested that developmental acceleration entailed a high metabolic cost: accelerating tadpoles experienced higher oxidative stress (Gomez-Mestre et al 2013 PLoS One) and they consumed a large fraction of their abdominal fat bodies compared to siblings kept under full water conditions (Kulkarni et al 2011 J Evol Biol). Thus, we hypothesized that *Scaphiopus* would have high, constitutive SMR to match its constitutively high basal CORT content. We now include this explanation in the text (line 143-146). The high constitutive SMR we found in *Scaphiopus* is consistent with the high constitutive CORT content, as well as lack of abdominal fat in *Scaphiopus* because exogenous CORT reduces abdominal fat bodies in tadpoles. Even blocking TH synthesis and thus metamorphosis using methimazole excluded abdominal fat accumulation in *Scaphiopus* but not in *Spea* (Kulkarni et al 2011 J Evol Biol), suggesting continuous high basal CORT levels that may promote lipid catabolism and preclude lipid storage. Thus, measuring SMR provides physiological functional relevance of our data on constitutively increased basal CORT content in *Scaphiopus*.

8) For the reasons outlined above, I'm not sure I agree with the authors' claim that their “current physiological results and our previous morphological results combine to provide the strongest evidence to date showing how genetic accommodation affecting developmental plasticity may have given rise to evolutionary differences among species in divergent environments.” However, whether one agrees with this statement or not, it is always best to let data speak for itself. Thus, I would remove such statements from the paper.

Response: It is always tricky to find a balance between scientific excitement plus generalization of the findings versus avoiding overstatements. We have toned down statements like the one pointed out by the reviewer throughout the MS while trying to explain the implications we think the results have on our understanding of genetic accommodation. For instance (Lines 154-158): “The interspecific differences we found in endocrine regulation of development rate and larval period plasticity and our previous results on linked morphological traits combine to lend strong support that genetic accommodation on ancestral developmental plasticity has given rise to evolutionary differences among species in divergent environments ”.

Minor concerns and suggestions.

On line 33, the authors state that they “show support for genetic accommodation beyond phenotypic comparisons”. However, this study shows “phenotypic comparisons”; it's just at a different level of phenotype that the authors' earlier studies on this system.

Response: The reviewer rightly points out this issue, we compared endocrine features to add to the morphological features previously examined. We also note that in addition to phenotypic comparisons, we performed CORT manipulations using CORT and metyrapone treatments. The introduction was greatly reworked and this phrase is no longer there.

Line 38: Should say, “spadefoot toad species”

Response: Changed.

Line 66: I would be careful about saying that Scaphiopus has “little or no fat”; what you’re really talking about here are abdominal fat bodies. In addition to these fat bodies, frogs (including spadefoots) store fat in their liver as well as subcutaneously.

Response: Good point, the sentence was incorrect as it was. We changed it to: ‘*and little or no abdominal fat bodies*’.

Line 69: CORT should be defined here

Response: Corrected.

Line 163: What do you mean by the phrase “genuine mechanism”? I would remove the word “genuine” as this may be construed by some to suggest that the mechanisms that other researchers have identified are not “genuine.”

Response: Removed.

Figures generally: The figures were hard for me to read in the 8 x 10-inch format at which I printed them out, and the authors should be aware of this. Also, I’m not sure that the box plots need to be shown in different colors to represent the different species – doing so just adds to the overall “noise” in the figures.

Response: We increased the font size to be as large as would reasonably fit. Also, the different species should be represented differently somehow because we wanted to make it as easy as possible to distinguish among species within a graph and compare within species across graphs. We kept the colors because the alternative would be shades of grey, which did not seem to reduce the “noise”, and lack of colors was not possible for Fig. 2 which required some way of demarcating species in the bars.

Figure 1 (B,C,D): It would be useful to see the actual values somewhere. Generally, for analyses, I’m curious as to why the authors did not perform a multiple comparisons test across all groups (in addition to, or instead of, doing pairwise tests within species). This seems like an unorthodox approach to me. I’m also concerned about the amount of variation in the “canalized” species in D. Of course, if there’s a lot of variation, you would not expect to find a significant difference in mean. Probably not a death stroke, but I think it’s important to recognize this when discussing the paper’s significance.

Response: We can put the actual values in a supplementary table for all the graphs, if desired; putting them in the figures would make them too crowded. We conducted general models first to test the effect of species and then stage or experimental treatment and their interaction. Then we conducted posthoc tests within species. Multiple comparisons across all groups would have unnecessarily inflated the number of posthoc tests and would have been hard to interpret (e.g. TH differences between Scaphiopus at 38 Gosner stage vs. Pelobates at 42 Gosner stage). We have clarified this in the statistical analyses section and provide a supplementary table with the overall models for each experiment.

Figure 2: It doesn’t say what conditions these data were taken from: high or low water?

Response: Larvae were raised in high water (50-60 cm) conditions for stage analyses. This

information was added to the figure legend.

Figure 3: I would like more explanation about the metyrapone treatment. If it is supposed to prevent synthesis of CORT, then why wouldn't it do that in *S. couchii*?

Response: In short, it is not known why CORT levels were not reduced by metyrapone treatment in *Scaphiopus*, but metyrapone is not always effective at reducing CORT levels in tadpoles. We now added the following text (line 174-179): "studies by others show that non-toxic doses of metyrapone are not always effective at significantly reducing CORT content, but expected CORT-related morphological and physiological effects may still be observed. Thus, our experiments show that SMR is regulated by environmentally altered CORT content at least in *P. cultripes* and *S. multiplicata*, and the high and invariant SMR in *S. couchii* is another example of a pattern of phenotypes among species consistent with genetic accommodation."

Line 310: Since the *P. cultripes* tadpoles were wild caught, how can the authors be certain that the developmental rate "decision" of their *P. cultripes* tadpoles wasn't influenced by their early life experience/environment?

Response: This possibility cannot be ruled out completely, but *P. cultripes* tadpoles in this experiment showed a very similar developmental rate and capacity for acceleration in response to decreased water level to those observed over the years in previous publications for this species, which were raised from eggs in the laboratory.

Line 327: "The sample size varied between 8 and 10 for each species." Are these families, tadpoles? Multiple populations? The answer to this question can be important because the degree of plasticity can vary at all these levels.

Response: We have changed the text to: "Each species is represented by at least three different clutches in each experiment, and specific sample sizes are provided in the figure legends" (line 397-399). It is true that populations, families and even siblings will vary in all the variables studied. Quantifying variation at those levels was beyond the scope of the current study, and thus interpretation of our data needs to take this potential limitation into account.

Line 330-331: What was the volume of the 100nM CORT and 25nM metyrapone used?

Response: Concentrations were adjusted from stock solutions so that 100 uL were added in all cases, whether control treatment (just ethanol) or hormonal treatments.

Line 336-337,343-344: "large stock tanks" What is this referring to?

Response: We changed this to "large stock tanks (60 cm H x 60 cm W x 120 cm L) filled 3/4 full with water".

Lines 368-371: Was the mass of each sample at least the same? Wouldn't it have been better to do some size-adjusted measurement? Or the same volume of homogenate?

Response: We compared CORT extracted from whole body homogenates (with intestines removed), but the tadpoles vary greatly in size among species and even within species between high and low water treatments. So, to make comparisons among species, all hormone measurements are given on a per mass unit. The volume of the homogenate used in the CORT extraction and RIA procedure was

approximately the same, where a single *Pelobates* tadpole could be used for the two hormones but 2-3 *Scaphiopus* tadpoles had to be pooled and homogenized together to measure both hormones.

The figures look like it is picograms of hormone per milligram of tissue. So is it higher amounts of hormone or lower amounts of tissue driving the pattern?

Response: The units of picograms hormone per milligram tissue convey tissue content of hormone (how much hormone per mass unit of tissue), which is the best way to compare amount of hormone across samples to avoid the problem that big tadpoles will have more hormone than small tadpoles all else being equal. Here we report that hormone content varies across species, stages, and treatments per mass unit.

Reviewers' comments:

Reviewer #1 (Remarks to the Author):

The authors have addressed most of my concerns satisfactorily and the paper is improved, but I still have four concerns:

1. For the message it has to convey, the writing in the introduction is still at times dense, opaque and indirect. I think the findings are very straightforward and the underlying hypothesis and interpretation can and should be explained in an equally clear manner, especially since this journal aims to reach a broad audience and disseminate findings of wide-reaching significance.

For example, the revised sentence "We found that ancestral mechanisms regulating developmental responses to environmental conditions characterized by a high level of larval period plasticity evolved in descendant species, giving rise to a canalized rapid developmental rate" took me at least two readings to get the point, and I have already reviewed the manuscript and am familiar with the underlying phylogenetic and ecological contexts. Ask some nonspecialists to read it and ask them not if they get the point but if it could be explained more clearly.

2. The authors are still not entirely clear about the significance of their findings in an evolutionary sense, and what exactly is the insight gained. It is already well established that genetic accommodation/assimilation (see my point 4) can occur when one population ceases to experience the environmental variability that previously maintained the developmental plasticity. The evidence of this is typically a difference between closely related species, i.e. one species still exhibits the plasticity and another that is restricted to a less variable environment is fixed for a trait that is adaptive for the environmental parameter that is no longer variable. In this case, short larval period is fixed for the species that lives mainly in short-lived larval habitats. This study goes on to provide convincing evidence that the loss of plasticity is due to the underlying neuroendocrine activity becoming constitutively maxed out, so to speak. Given that tadpole developmental rate is well known to be regulated by tadpole neuroendocrine activity, this is not really very surprising, though it is of course important to show that this is the case.

The question that persists in my head is What more does this tell us about "how mechanisms of trait regulation evolve during genetic accommodation" and "why/how lineages differ in phenotype". Emphasizing that the results address the "how/why" of differences between lineages and show a mechanistic link in trait evolution at both micro- and macroevolutionary scales doesn't really say much since this is all implied by the term genetic accommodation in the first place. Also, the terms "lineage" and "lineage within species" seem deliberately vague.

My understanding is that genetic accommodation (and assimilation) are intraspecific evolutionary mechanisms, so by using the term the authors are acknowledging that the mechanism happened before the descendent species became its own species. This brings us

back to my original concern about what more can be said about its role in speciation, or phenotypic evolution more generally, and directions for future research. Since the authors have spent a lot of their careers researching the hormonal and molecular bases of tadpole development, can they not shed some insight into how regulatory mechanisms might be altered to max out a tadpole developmental rate.

3. Why is genetic accommodation referred to as predictive or having expectations, e.g., “match the expectations of genetic accommodation” and “divergent reaction norms among species are an expected pattern in evolution by genetic accommodation”. As I point out above, “divergent reaction norms among species”, is the evidence for why we suspect the occurrence of genetic accommodation or assimilation. In other words, without that divergence, we have no reason to infer its occurrence. To then go on and refer to the evidence as a prediction or expectation seems circular to me. Part of the problem is that genetic accommodation/assimilation can only be recognized after it happens on the basis of the evolutionary pattern that results, i.e., the divergent reaction norms.

4. Since the derived developmental trait, fast larval development, is constitutive or canalized, why are the authors not calling it a product of genetic assimilation? The literature dating back to its origin with Waddington generally uses the term genetic assimilation for formerly plastic phenotypes that become constitutively expressed or canalized. This is also explained on p. 148 of West-Eberhard’s definitive text, and p. 384 of Gilbert and Epel’s excellent review of the topic in *Ecological Developmental Biology*. The latter source emphasizes that accommodation and plasticity are actually “opposite sides of the same coin”: one refers to the stabilization of plasticity and the other to its loss.

Chrs Rose

Reviewer #3 (Remarks to the Author):

This paper is a revision of an earlier paper that seeks to clarify the endocrine mechanisms that underlie genetic accommodation of developmental rate (and correlated traits) in spadefoot toads. As I noted in my earlier review, there are actually few clear-cut examples of genetic accommodation in natural populations and fewer still known to account for phenotypic diversity among species. Therefore, this paper has the potential to make an important contribution to the literature on the underlying mechanisms of genetic accommodation.

I found that revised version of the paper to be an improvement over the first submission. In particular, the authors responses were thoughtful and showed that they clearly thought hard about the critiques raised of their earlier version. I was especially pleased to see that the authors have decided to reframe their Introduction by focusing it on establishing the underlying endocrine mechanisms of genetic accommodation in this system (and removing the material on a shared mechanism being a required assumption of genetic accommodation).

At the same time, I'm concerned that some of my earlier comments (and, perhaps, those of the other reviewers as well) have not actually been addressed in the manuscript itself.

Specifically, there is no indication that my original comment #2 was addressed in the text (e.g., by saying how THIS endocrine change is consistent with genetic accommodation but some other change, such as more hormone receptors, is not consistent). Similarly, my original comments #5 & #6 are not addressed in the manuscript itself (my original comments are listed below the line demarcated by "+++++"). In my view, all of these should be addressed in the text, as they are likely concerns that will be raised by other readers.

In addition to these concerns not being addressed in the text, I have concerns with the authors' response to my original comment #6 below. In addressing this concern in their cover letter (but, again, not in the text!), the authors maintain that "Thus, these species do differ in where their eggs end up (with some overlap), such that the tadpoles in the two species do experience on average different hydroperiods." I have two concerns here. First, in my field experience (and from talking to other researchers who have had lots of experience studying the natural history of these two species), while *Scaphiopus couchii* do sometimes breed in very ephemeral ponds that *Spea* cannot withstand, *Scaphiopus couchii* is nearly always present in the same ponds as *Spea multiplicata*. Thus, it is hard to say that the two species are indeed experiencing divergent environments, since they are generally present together.

Second, I recently came across a 2014 PLoS One paper co-authored by one of the authors on this paper ("Evolution of Rapid Development in Spadefoot Toads Is Unrelated to Arid Environments," by Cen Zeng, Ivan Gomez-Mestre, John J. Wiens), which found "no significant relationships between life-history variables and precipitation or aridity levels where these species occur. Instead, rapid development in pelobatoids is strongly related to their small genome sizes and to phylogeny." So, it sounds as if selection has not acted on this system in the way that the authors maintained it has in the present paper (selection is important here, because genetic accommodation is, by definition, a evolutionary response to selection). At the very least, this point needs to be clarified IN THE TEXT (and not just in the cover letter).

+++++

Concerns from first review not addressed in the text of the revision:

2) Second, and related to the previous point above, the fact that (in the present case) the same endocrine mechanism underlies both developmental plasticity and interspecific phenotypic differences is not at all surprising. The authors state that they sought to establish whether (lines 44-46) "a commonality of developmental endocrine mechanisms underlies both developmental plasticity and interspecific phenotypic differences," but on lines 70-72, they state that "developmental acceleration in response to pond drying in spadefoot toads is, as in all other anurans, largely dependent upon increased levels of thyroid hormone (TH) and corticosterone (CORT)." Thus, these hormones were GUARANTEED to regulate developmental rate both within versus between species.

In other words, the underlying mechanisms MUST be the same within and between species. So, in a sense, the outcome was not surprising nor was it unanticipated. Again, this weakens the authors' assertion that a key assumption of genetic accommodation is that a commonality of developmental endocrine mechanisms must underlie both developmental plasticity and interspecific phenotypic differences – this will likely be true in MANY species at the broad level of endocrine mechanisms, where there is a limited number of hormones that could plausibly mediate such responses!

5) As the authors are aware, *Spea multiplicata* produces alternative larval ecomorphs (as part of a polyphenism) that also differ in developmental rate: a slow-developing omnivore morph and a more rapidly-developing carnivore morph. Could the existence of this omnivore-carnivore polyphenism explain why much greater levels of plasticity were observed in *Spea* than in *Scaphiopus* (because the former included both morphs)? If the data presented in this paper are from omnivores only, how does the exclusion of carnivores affect the results, since most natural populations would actually produce both morphs? It seems that excluding carnivores would give one a greatly biased representation of developmental rate (and the correlated traits) in *Spea*.

6) Contrary to what the authors claim here, *Spea multiplicata* and *Scaphiopus couchii* do NOT appear to be experiencing "divergent environments", at least not divergent hydroperiod environments, which is implied by the authors. The two species have nearly completely overlapping geographical distributions (in the U.S. at least), and, indeed, their tadpoles often co-occur in the SAME ponds. Why, then, have they diverged so much in developmental rate (and associated traits)? This seems to be a major gap in the authors' story on this system. Some other agent(s) of selection must be in play here.

Responses to reviewers

We very much appreciate the reviewers' continued interest in our manuscript. We have again made our best effort to be clear in our writing based on reviewer comments by overhauling the abstract, introduction, and discussion sections and much of the results, indicated in red from the previous version. Despite many changes to the text, our original interpretation of the results and conclusions are not changed. Also, all reviewer comments have been addressed in the text of the manuscript. We are excited about this version because of the much improved presentation of material, which makes a novel and significant contribution to understand genetic accommodation and species phenotypic differences.

Reviewers' comments:

Reviewer #1 (Remarks to the Author):

The authors have addressed most of my concerns satisfactorily and the paper is improved, but I still have four concerns:

1. For the message it has to convey, the writing in the introduction is still at times dense, opaque and indirect. I think the findings are very straightforward and the underlying hypothesis and interpretation can and should be explained in an equally clear manner, especially since this journal aims to reach a broad audience and disseminate findings of wide-reaching significance.

For example, the revised sentence “We found that ancestral mechanisms regulating developmental responses to environmental conditions characterized by a high level of larval period plasticity evolved in descendant species, giving rise to a canalized rapid developmental rate” took me at least two readings to get the point, and I have already reviewed the manuscript and am familiar with the underlying phylogenetic and ecological contexts. Ask some nonspecialists to read it and ask them not if they get the point but if it could be explained more clearly.

Response: Thank you for continuing to request clear writing, as it is our goal as well. We have overhauled much of the text including the abstract, Introduction, and Discussion. Our intended message hasn't changed, but we have done our best to make the meaning clear.

2. The authors are still not entirely clear about the significance of their findings in an evolutionary sense, and what exactly is the insight gained. It is already well established that genetic accommodation/assimilation (see my point 4) can occur when one population ceases to experience the environmental variability that previously maintained the developmental plasticity. The evidence of this is typically a difference between closely related species, i.e. one species still exhibits the plasticity and another that is restricted to a less variable environment is fixed for a trait that is adaptive for the

environmental parameter that is no longer variable. In this case, short larval period is fixed for the species that lives mainly in short-lived larval habitats. This study goes on to provide convincing evidence that the loss of plasticity is due to the underlying neuroendocrine activity becoming constitutively maxed out, so to speak. Given that tadpole developmental rate is well known to be regulated by tadpole neuroendocrine activity, this is not really very surprising, though it is of course important to show that this is the case.

The question that persists in my head is What more does this tell us about “how mechanisms of trait regulation evolve during genetic accommodation” and “why/how lineages differ in phenotype”. Emphasizing that the results address the “how/why” of differences between lineages and show a mechanistic link in trait evolution at both micro- and macroevolutionary scales doesn’t really say much since this is all implied by the term genetic accommodation in the first place. Also, the terms “lineage” and “lineage within species” seem deliberately vague.

My understanding is that genetic accommodation (and assimilation) are intraspecific evolutionary mechanisms, so by using the term the authors are acknowledging that the mechanism happened before the descendent species became its own species. This brings us back to my original concern about what more can be said about its role in speciation, or phenotypic evolution more generally, and directions for future research. Since the authors have spent a lot of their careers researching the hormonal and molecular bases of tadpole development, can they not shed some insight into how regulatory mechanisms might be altered to max out a tadpole developmental rate.

Response: The developmental/genetic/endocrine mechanisms underlying genetic accommodation are not well understood, especially among species. The most recent review on genetic accommodation compiled 146 studies where lineages (populations or species) had apparently evolved through genetic accommodation (Schlichting and Wund 2014). Out of those 146 studies only 12 focused on differences among species, while the rest focused on evolution of plasticity among populations. Two of those 12 studies come from our own research group. The remaining 10 articles describe possible cases among related or unrelated species of genetic accommodation, dealing exclusively with behavioral or morphological comparisons rather than developmental genetic mechanisms. Thus, we believe that showing the mechanistic basis at the endocrine level for explaining phenotypic differences among species is the core significance and novelty of our data. We removed the terms micro- and macroevolution as they may imply something different than what our data address and replace the last sentence of the abstract with: "Our findings support that the atypically short and canalized development of *S. couchii* evolved by genetic accommodation of endocrine pathways controlling metamorphosis, showing how phenotypic plasticity within species may evolve into trait variation among species." Also, we have changed "lineage" to "lineage (population or species)" and "lineages within species" is no longer used.

In the last paragraph of the Discussion we acknowledge the significance of the neuroendocrine signaling: "Because of their pivotal roles in controlling larval period and developmental plasticity, variation in TH and CORT production and signaling was a key

factor in the evolution of larval period and plasticity in spadefoot toads." However, the key issue is *how* endocrine regulation changed beyond involving these two hormones. We expanded on the neuroendocrine signaling in the body of the Discussion: "Endocrine regulation of tadpole metamorphosis is complex, and so numerous changes in the endocrine system could have accounted for the short larval period of *S. couchii*. However, the specific endocrine regulation we observed underlying canalized development in *S. couchii* compared to its relatives implicates evolution by genetic accommodation. In particular, the endocrine changes and morphological consequences that accompany accelerated development in *P. cultripes*, such as increased TH and CORT, smaller body size at metamorphosis, shorter hind limb length, and reduced size of the abdominal fat bodies, are constitutive features of the faster developing, derived species (i.e., *S. couchii*)^{12,14,15}. It appears that, since the last common ancestor of *S. couchii* and the other species, selection stabilized the short larval period phenotype that was previously obtained only by environmental induction."

In the absence of such mechanistic analyses, other hypotheses about how species differences come about could not be ruled out. As we explain in the Discussion: "A hypothetical alternative evolutionary pathway to achieve short larval periods is that *S. couchii* may have evolved a short larval period via evolution of rapid tissue transformation with reduced dependence on TH, such that none of the other changes associated with accelerated metamorphosis in plastic species, involving metabolism, limbs, gonads, and fat bodies, would necessarily have been obtained." Thus, our analysis of neuroendocrine mechanisms of development and plasticity was critical to our conclusions and provide rare evidence that genetic accommodation may account for phenotypic differences among spadefoot species.

It is true that genetic accommodation is an intraspecific process. However, it is not necessary for our thesis that the relevant genetic accommodation happened within the ancestral species before *S. couchii* became its own species. As we explain in the Discussion: "We envision the following evolutionary scenario relating plasticity to phenotypic divergence among spadefoot toad species. First, Old and New World spadefoot toad clades (*Pelobates* vs *Spea/Scaphiopus*) diverged at least by the Early Cretaceous²⁶⁻²⁸, perhaps due to the formation of the Atlantic Ocean 150 Ma. Plasticity in larval period was present before this split, as it is a common feature of amphibian development and was retained in both lineages after the split¹⁴. Then, ancestors of *Scaphiopus* and *Spea* gradually experienced increased pond ephemerality compared with ancestors of *Pelobates*, but developmental plasticity of their larvae would have enabled them to persist. Because the timing of metamorphosis depends on endocrine signaling, continued selection for short larval periods to survive ephemeral ponds in New World spadefoot toad ancestors resulted in genetic changes that altered pathways controlling hormone production and/or action."

We also indicate molecular mechanisms that have been identified that may account for the diverged larval period of *S. couchii*: "At a mechanistic level, the rapid development of *S. couchii* may be explained by the higher tissue content of TH as well as increased sensitivity to TH action, due to higher expression of thyroid hormone receptors^{10,29}. Also, reduced plasticity in larval period in *S. couchii* may be explained by the maintenance of high TH and CORT production at near climax levels throughout the larval period⁹."

3. Why is genetic accommodation referred to as predictive or having expectations, e.g., “match the expectations of genetic accommodation “ and “divergent reaction norms among species are an expected pattern in evolution by genetic accommodation”. As I point out above, “divergent reaction norms among species”, is the evidence for why we suspect the occurrence of genetic accommodation or assimilation. In other words, without that divergence, we have no reason to infer its occurrence. To then go on and refer to the evidence as a prediction or expectation seems circular to me. Part of the problem is that genetic accommodation/assimilation can only be recognized after it happens on the basis of the evolutionary pattern that results, i.e., the divergent reaction norms.

Response: We have removed that kind of argument from the text. We agree that we cannot claim the endocrine and phenotypic differences among species points to genetic accommodation and then turn around to say how species differences in phenotype are explained by genetic accommodation. We also agree that genetic accommodation doesn't have any particular expected patterns of phenotypic differences among species. We rephrased these statements to say that spadefoot phenotypic differences within and among species "are consistent with" or "provide evidence for" evolution by genetic accommodation throughout the manuscript.

4. Since the derived developmental trait, fast larval development, is constitutive or canalized, why are the authors not calling it a product of genetic assimilation? The literature dating back to its origin with Waddington generally uses the term genetic assimilation for formerly plastic phenotypes that become constitutively expressed or canalized. This is also explained on p. 148 of West-Eberhard's definitive text, and p. 384 of Gilbert and Epel's excellent review of the topic in Ecological Developmental Biology. The latter source emphasizes that accommodation and plasticity are actually “opposite sides of the same coin”: one refers to the stabilization of plasticity and the other to its loss.

Response: We have adopted the view that genetic assimilation is an extreme case of the more general concept of genetic accommodation, as used by Crispo 2007 Evolution; Schlichting & Wund 2014 and proposed in West-Eberhard (2003, p. 157, she seems to contradict herself when compared to what she said on p. 148). Also, in genetic assimilation, the trait is supposed to have evolved to be completely free of environmental influence and under strict genetic control, which we did not see in *S. couchii*. Also, although *Scaphiopus* has reduced plasticity to a large extent compared to *Pelobates*, it still retains some degree of plasticity and in fact it was described as an example of developmental plasticity in previous literature (e.g. Newman 1988 Evolution; Newman 1989 Ecology). For these two reasons, we think that genetic accommodation should be an acceptable term here.

Reviewer #3 (Remarks to the Author):

This paper is a revision of an earlier paper that seeks to clarify the endocrine mechanisms that underlie genetic accommodation of developmental rate (and correlated traits) in spadefoot toads. As I noted in my earlier review, there are actually few clear-cut examples of genetic accommodation in natural populations and fewer still known to account for phenotypic diversity among species. Therefore, this paper has the potential to make an important contribution to the literature on the underlying mechanisms of genetic accommodation.

I found that revised version of the paper to be an improvement over the first submission. In particular, the authors responses were thoughtful and showed that they clearly thought hard about the critiques raised of their earlier version. I was especially pleased to see that the authors have decided to reframe their Introduction by focusing it on establishing the underlying endocrine mechanisms of genetic accommodation in this system (and removing the material on a shared mechanism being a required assumption of genetic accommodation).

At the same time, I'm concerned that some of my earlier comments (and, perhaps, those of the other reviewers as well) have not actually been addressed in the manuscript itself.

Specifically, there is no indication that my original comment #2 was addressed in the text (e.g., by saying how THIS endocrine change is consistent with genetic accommodation but some other change, such as more hormone receptors, is not consistent). Similarly, my original comments #5 & #6 are not addressed in the manuscript itself (my original comments are listed below the line demarcated by "+++++"). In my view, all of these should be addressed in the text, as they are likely concerns that will be raised by other readers.

Response: We now directly address all comments in text, see below.

In addition to these concerns not being addressed in the text, I have concerns with the authors' response to my original comment #6 below. In addressing this concern in their cover letter (but, again, not in the text!), the authors maintain that "Thus, these species do differ in where their eggs end up (with some overlap), such that the tadpoles in the two species do experience on average different hydroperiods." I have two concerns here. First, in my field experience (and from talking to other researchers who have had lots of experience studying the natural history of these two species), while *Scaphiopus couchii* do sometimes breed in very ephemeral ponds that *Spea* cannot withstand, *Scaphiopus couchii* is nearly always present in the same ponds as *Spea multiplicata*. Thus, it is hard to say that the two species are indeed experiencing divergent environments, since they are generally present together.

Response: We now address this in the Discussion: "The strength of selection in favor of reduced larval period likely differs among species. That is, *P. cultripes*, whose larval

period ranges from 93 to 186 days, breeds in long lasting temporary ponds that eventually dry up in summer; whereas, *S. couchii*, whose larval period ranges 7 to 30 days lays eggs in ephemeral desert pools that often dry in less than 2 weeks. Furthermore, despite substantial overlap in breeding ponds, *S. couchii* often chooses pools that are too ephemeral for *S. multiplicata*, whose larval period ranges from 12 to 40 days. The exploitation of extremely ephemeral pools by *S. couchii* represents recurring episodes of stronger selection for rapid development than that experienced by *S. multiplicata*." This argument should be sufficient at least for *S. couchii* vs. *P. cultripes* if not also for *S. couchii* vs *S. multiplicata*.

Second, I recently came across a 2014 PLoS One paper co-authored by one of the authors on this paper ("Evolution of Rapid Development in Spadefoot Toads Is Unrelated to Arid Environments," by Cen Zeng, Ivan Gomez-Mestre, John J. Wiens), which found "no significant relationships between life-history variables and precipitation or aridity levels where these species occur. Instead, rapid development in pelobatoids is strongly related to their small genome sizes and to phylogeny." So, it sounds as if selection has not acted on this system in the way that the authors maintained it has in the present paper (selection is important here, because genetic accommodation is, by definition, a evolutionary response to selection). At the very least, this point needs to be clarified IN THE TEXT (and not just in the cover letter).

Response: It appears true that aridity may not account for selection for short larval periods, but rather, as we now state in the Discussion, "Pond duration, rather than terrain aridity, is the most likely environmental factor that has driven the evolution of developmental rate in these species." This situation does not change our conclusions regarding developmental plasticity and phenotypic divergence among species. Regarding genome size, we now state in the Discussion: "Genome size differences correlate with differences in larval period among species, such that species with faster development rate had reduced genome sizes¹⁹. To our knowledge, there is no known association between the level of plasticity and genome size across species, and this pattern is likely a by-product of selection to achieve faster development via more rapid cell division, also found in insects³⁰."

+++++

Concerns from first review not addressed in the text of the revision:

2) Second, and related to the previous point above, the fact that (in the present case) the same endocrine mechanism underlies both developmental plasticity and interspecific phenotypic differences is not at all surprising. The authors state that they sought to establish whether (lines 44-46) "a commonality of developmental endocrine mechanisms underlies both developmental plasticity and interspecific phenotypic differences," but on lines 70-72, they state that "developmental acceleration in response to pond drying in spadefoot toads is, as in all other anurans, largely dependent upon increased levels of thyroid hormone (TH) and corticosterone (CORT)." Thus, these hormones were GUARANTEED to regulate developmental rate both within versus

between species.

In other words, the underlying mechanisms MUST be the same within and between species. So, in a sense, the outcome was not surprising nor was it unanticipated. Again, this weakens the authors' assertion that a key assumption of genetic accommodation is that a commonality of developmental endocrine mechanisms must underlie both developmental plasticity and interspecific phenotypic differences – this will likely be true in MANY species at the broad level of endocrine mechanisms, where there is a limited number of hormones that could plausibly mediate such responses!

Response: This issue was also a concern of the first reviewer. In the last paragraph of the Discussion we acknowledge the significance of the neuroendocrine signaling: "Because of their pivotal roles in controlling larval period and developmental plasticity, variation in TH and CORT production and signaling was a key factor in the evolution of larval period and plasticity in spadefoot toads." However, the key issue is how endocrine regulation changed beyond involving these two hormones. We expand on the neuroendocrine signaling in the body of the Discussion: "Endocrine regulation of tadpole metamorphosis is complex, and so numerous changes in the endocrine system could have accounted for the short larval period of *S. couchii*. However, the specific endocrine regulation we observed underlying canalized development in *S. couchii* compared to its relatives implicates evolution by genetic accommodation. In particular, the endocrine changes and morphological consequences that accompany accelerated development in *P. cultripes*, such as increased TH and CORT, smaller body size at metamorphosis, shorter hind limb length, and reduced size of the abdominal fat bodies, are constitutive features of the faster developing, derived species (i.e., *S. couchii*) ^{12,14,15}. It appears that, since the last common ancestor of *S. couchii* and the other species, selection stabilized the short larval period phenotype that was previously obtained only by environmental induction." In the absence of such mechanistic analyses, other hypotheses about how species differences come about could not be ruled out. As we explain in the Discussion: "A hypothetical alternative evolutionary pathway to achieve short larval periods is that *S. couchii* may have evolved a short larval period via evolution of rapid tissue transformation with reduced dependence on TH, such that none of the other changes associated with accelerated metamorphosis in plastic species, involving metabolism, limbs, gonads, and fat bodies, would necessarily have been obtained." Thus, our analysis of neuroendocrine mechanisms of development and plasticity was critical to our conclusions and provide rare evidence that genetic accommodation may account for phenotypic differences among spadefoot species.

5) As the authors are aware, *Spea multiplicata* produces alternative larval ecomorphs (as part of a polyphenism) that also differ in developmental rate: a slow-developing omnivore morph and a more rapidly-developing carnivore morph. Could the existence of this omnivore-carnivore polyphenism explain why much greater levels of plasticity were observed in *Spea* than in *Scaphiopus* (because the former included both morphs)? If the data presented in this paper are from omnivores only, how does the exclusion of carnivores affect the results, since most natural populations would actually produce both

morphs? It seems that excluding carnivores would give one a greatly biased representation of developmental rate (and the correlated traits) in *Spea*.

Response: We now include the following in the Discussion: "*S. multiplicata* can produce carnivorous tadpoles in nature in response to appropriate environmental cues, and these develop faster than omnivore morphs. However, omnivore morphs seem to be produced as the default in nature, and carnivore morphs are exceedingly rare under laboratory rearing conditions (<1 per 1000, even when trying to produce them by altering tadpole density and using anostracan shrimp, personal obs.). Neither *Pelobates* nor *Scaphiopus* are known to produce carnivore morphs. Thus, because our experiments depended on identical laboratory conditions, carnivore morphs were not observed in high or low water and thus did not impact the developmental rate or developmental plasticity observed in *Spea*. "

6) Contrary to what the authors claim here, *Spea multiplicata* and *Scaphiopus couchii* do NOT appear to be experiencing "divergent environments", at least not divergent hydroperiod environments, which is implied by the authors. The two species have nearly completely overlapping geographical distributions (in the U.S. at least), and, indeed, their tadpoles often co-occur in the SAME ponds. Why, then, have they diverged so much in developmental rate (and associated traits)? This seems to be a major gap in the authors' story on this system. Some other agent(s) of selection must be in play here.

Response: See our response to Comment #6 above.

REVIEWERS' COMMENTS:

Reviewer #1 (Remarks to the Author):

The paper has been dramatically improved through better use of terminology, the inclusion of discussion on possible underlying mechanisms and other potentially relevant factors (genome size and carnivore morphs in one species), and a more clear-cut presentation of the hypothesis being tested. Though the writing style has also been improved, I still see some easily fixable problems that have to do with excessive compression of language and others that have to do with the use of terms and interpretation of results:

Line 31 We found that *Pelobates cultripes* and *Spea multiplicata* accelerate metamorphosis in response to pond drying, accompanied by increased standard metabolic rate (SMR) and elevated whole-body content of thyroid hormone

What is accompanied by the increased metabolic rate and elevated hormone levels ...the two species, or the metamorphosis, or the acceleration of metamorphosis by the two species? I understand it is the latter, but why use such an ugly and vague construction, even if journals permit it, when it is entirely unnecessarily. Does "Currey passes the ball, accompanied by a smile to the fans" sound right to you?

Line 36 Saying the highest values of the three traits are minimally affected by pond drying is incorrect. You mean that the trait values in *S. couchi* are the highest for the three species, and are minimally (or least) affected by pond drying.

Line 86, I do not understand an "environmentally induced metamorphosis". Do you mean one that has been accelerated by environmental factors, or one that is not induced by exogenous hormones, which used to be called a spontaneous metamorphosis in the older literature?

Line 102, 113, 195: Isn't "genetic accommodation of developmental plasticity" redundant? Doesn't any discussion of genetic accommodation always apply to developmental plasticity, regardless of whether you are talking about traits, regulation of traits, frequency of traits, etc. Also, after rereading the West-Eberhard book and Crispo paper, I am Ok with the term genetic accommodation being used instead of genetic assimilation, though I still think a clear and complete definition of genetic accommodations remains elusive, and genetic assimilation is still a valid term and has a very precise and hence very useful definition.

Line 144 "Like TH, *S. couchii* had higher whole-body CORT..." is an example of a dangling modifier. *S. couchi* is not like TH.

Line 251 I am curious to know more about the reduced plasticity in *S. couchi*. In their response to my previous review, the authors state "Also, although *Scaphiopus* has reduced plasticity to a large extent compared to *Pelobates*, it still retains some degree of plasticity and in fact it was described as an example of developmental plasticity in previous literature (e.g. Newman 1988 Evolution; Newman 1989 Ecology). I think this is something that should

be explained in the text, since it is relevant and is apparently why the authors are deciding not to recognize this as an example of genetic assimilation.

Line 273 However, omnivore morphs seem to be produced as the default in nature, and carnivore morphs are exceedingly rare under laboratory rearing conditions (<1 per 1000, even when trying to produce them by altering tadpole density and using anostracan shrimp, personal obs.).

I appreciated this inclusion since this has always been a mystery to me (why no one has unraveled the exact basis of developmental basis of this polyphenism).

Line 286 "canalization of ancestral plasticity". Development can be canalized, but I do not think that plasticity can be canalized. Plasticity is only lost or reduced, possibly as a result of canalization.

Chris Rose

We agree with the criticisms and provide our corrections in the responses below.

Reviewer #1 (Remarks to the Author):

The paper has been dramatically improved through better use of terminology, the inclusion of discussion on possible underlying mechanisms and other potentially relevant factors (genome size and carnivore morphs in one species), and a more clear-cut presentation of the hypothesis being tested. Though the writing style has also been improved, I still see some easily fixable problems that have to do with excessive compression of language and others that have to do with the use of terms and interpretation of results:

Line 31 We found that *Pelobates cultripes* and *Spea multiplicata* accelerate metamorphosis in response to pond drying, accompanied by increased standard metabolic rate (SMR) and elevated whole-body content of thyroid hormone

What is accompanied by the increased metabolic rate and elevated hormone levels ...the two species, or the metamorphosis, or the acceleration of metamorphosis by the two species? I understand it is the latter, but why use such an ugly and vague construction, even if journals permit it, when it is entirely unnecessarily. Does "Currey passes the ball, accompanied by a smile to the fans" sound right to you?

Response: We have modified the Editor's fix to "We find that, in response to pond drying, *Pelobates cultripes* and *Spea multiplicata* accelerate metamorphosis, increase standard metabolic rate (SMR), and elevate whole-body content of thyroid hormone (the primary morphogen controlling metamorphosis) and corticosterone (a stress hormone acting synergistically with thyroid hormone to accelerate metamorphosis)."

Line 36 Saying the highest values of the three traits are minimally affected by pond drying is incorrect. You mean that the trait values in *S. couchii* are the highest for the three species, and are minimally (or least) affected by pond drying.

Response: We have modified the sentence to "In contrast, *Scaphiopus couchii* has the shortest larval period, highest whole-body TH and CORT content, and highest SMR, and these trait values are least affected by pond drying among the three species."

Line 86, I do not understand an "environmentally induced metamorphosis". Do you mean one that has been accelerated by environmental factors, or one that is not induced by exogenous hormones, which used to be called a spontaneous metamorphosis in the older literature?

Response: We mean to refer to certain environmental conditions relative to more neutral conditions that induce an increased rate of metamorphosis via a hormonal response by the tadpole to the environment. Spontaneous metamorphosis doesn't capture this idea, and neither does "metamorphosis accelerated by environmental factors" which could refer to some passive effect like increased temperature that doesn't necessarily involve a response from the

organism. A term often used is "stress-induced" metamorphosis, which we now use here as well even though "stress" may be too strong a word.

Line 102, 113, 195: Isn't "genetic accommodation of developmental plasticity" redundant? Doesn't any discussion of genetic accommodation always apply to developmental plasticity, regardless of whether you are talking about traits, regulation of traits, frequency of traits, etc. Also, after rereading the West-Eberhard book and Crispo paper, I am Ok with the term genetic accommodation being used instead of genetic assimilation, though I still think a clear and complete definition of genetic accommodations remains elusive, and genetic assimilation is still a valid term and has a very precise and hence very useful definition.

Response: We agree that "genetic accommodation of developmental plasticity" is redundant. For line 195, we specified plasticity of timing of metamorphosis and so that is not redundant. Similarly for line 113, we specified the larval period but removed "developmental plasticity". For 102, we removed "developmental plasticity", and note that the context indicates larval period.

Line 144 "Like TH, *S. couchii* had higher whole-body CORT..." is an example of a dangling modifier. *S. couchii* is not like TH.

Response: We have changed the sentence to "As was the case for TH, *S. couchii* had higher whole-body CORT content...".

Line 251 I am curious to know more about the reduced plasticity in *S. couchii*. In their response to my previous review, the authors state "Also, although *Scaphiopus* has reduced plasticity to a large extent compared to *Pelobates*, it still retains some degree of plasticity and in fact it was described as an example of developmental plasticity in previous literature (e.g. Newman 1988 Evolution; Newman 1989 Ecology). I think this is something that should be explained in the text, since it is relevant and is apparently why the authors are deciding not to recognize this as an example of genetic assimilation.

Response: We added the following to the Introduction: "... even though *S. couchii* was described as a clear example of developmental plasticity in previous literature, this species has dramatically reduced plasticity compared to *P. cultripes*."

Line 273 However, omnivore morphs seem to be produced as the default in nature, and carnivore morphs are exceedingly rare under laboratory rearing conditions (<1 per 1000, even when trying to produce them by altering tadpole density and using anostracan shrimp, personal obs.).

I appreciated this inclusion since this has always been a mystery to me (why no one has unraveled the exact basis of developmental basis of this polyphenism).

Response: We modified this section to avoid "pers. obs" and add details to this issue: "With regard to the expression of the induced carnivorous morphology, *S. multiplicata* can produce carnivorous tadpoles in response to appropriate environmental cues, including anostrocan fairy shrimp and *Scaphiopus* tadpoles as food, but omnivore morphs seem to be produced as the default in nature. Even though carnivore morphs develop faster than omnivore morphs, survival through metamorphosis did not differ between carnivore and omnivore morphs in a pond drying experiment. In any case, carnivore morphs were not observed in our experiments and thus did not impact the developmental rate or plasticity we observed in *Spea*. "

Line 286 "canalization of ancestral plasticity". Development can be canalized, but I do not think that plasticity can be canalized. Plasticity is only lost or reduced, possibly as a result of canalization.

Response: We have replaced the phrase with "reduction in ancestral plasticity".